# All ERMs Can Fail in Stochastic Convex Optimization Lower Bounds in Linear Dimension

**Tal Burla** [1]   **Roi Livni** [2]

## Abstract

We study the sample complexity of the *best-case* Empirical Risk Minimizer in the setting of stochastic convex optimization. We show that there exists an instance in which the sample size is linear in the dimension, learning is possible, but the Empirical Risk Minimizer is likely to be *unique* and to *overfit*. This resolves an open question by Feldman. We also extend this to approximate ERMs.

Building on our construction we also show that (constrained) Gradient Descent potentially overfits when horizon and learning rate grow w.r.t sample size. Specifically we provide a novel generalization lower bound of $\Omega\left(\sqrt{\eta T/m^{1.5}}\right)$ for Gradient Descent, where $\eta$ is the learning rate, $T$ is the horizon and $m$ is the sample size. This narrows down, exponentially, the gap between the best known upper bound of $O(\eta T/m)$ and existing lower bounds from previous constructions.

## 1. Introduction

One of the central puzzles in contemporary learning theory is to understand how *overparameterized* learning is possible. In the overparameterized regime, learners can often perfectly fit the training data. On the one hand, this allows the learner to explore versatile, complex models. However, reasoning about generalization becomes difficult and standard concentration arguments become weak or non-uniform. Yet overparameterization is central to the success of modern learning systems. Modern Machine Learning builds on highly overcapacitated models, and optimization methods that explore loss landscapes that are seeded with overfitting

minima.

Several works attribute this success to *implicit bias* of the optimization algorithm (Neyshabur et al., 2014; Zhang et al., 2021). They postulate that from all fitting solutions, the algorithm converges to a statistically simpler subset. In contrast, other works suggest that the bias may be intrinsic to the loss landscape (Chiang et al., 2022; Valle-Perez et al., 2018; Buzaglo et al., 2024). These works suggest that for large models, a *typical* solution that fits the data may already generalize. To make this intuition precise, Chiang, Ni, Miller, Bansal, Geiping, Goldblum, and Goldstein (2022) proposed the following *thought experiment*:

> Suppose we pick a random solution that fits the data. How would its generalization behavior look?

While such an experiment is infeasible to carry out directly on real-world practical problems, the authors used intermediary datasets and proxy methodologies to conduct experiments that shed light on this question.

Another approach to tackle this question is to formalize the thought experiment within a theoretical setting that already captures the core elements of the problem. A natural and well-studied framework for this purpose is Stochastic Convex Optimization (SCO). The question might seem too basic, due to the simplicity of the loss landscape. But as it turns out it already contains all the necessary ingredients to make the problem interesting. First, existing algorithms that are prototypical for existing learning methods, such as Stochastic Gradient Descent (SGD) (Robbins & Monro, 1951; Hazan, 2016), come with provable guarantees and learn with sample size that is *dimension independent*. Second, it is known (Feldman, 2016; Shalev-Shwartz et al., 2009) that, unless the number of examples is proportional to the dimension of the parameter space, there are bad overfitting minima. In fact, even natural algorithms such as GD can overfit (Livni, 2024).

To summarize, the setting is an overparameterized setting in the following sense: Without dimension dependent sample size there are overfitting solutions, but there are learning algorithms that don't require dimension-dependent sample size. As such, it is an ideal setting to investigate the role of the loss landscape, the behavior of generic empirical risk

[1]Blavatnik School of Computer Science and AI, Tel Aviv University, Tel Aviv, Israel [2]School of Electrical and Computer Engineering, Tel Aviv University, Tel Aviv, Israel. Correspondence to: Tal Burla <talburla@mail.tau.ac.il>, Roi Livni <rlivni@tauex.tau.ac.il>.

*Proceedings of the 43rd International Conference on Machine Learning*, Seoul, South Korea. PMLR 306, 2026. Copyright 2026 by the author(s).

minimizers, and to analyze a Guess & Check algorithm that randomly chooses a solution that fits the data.

Certain intermediate answers exist in the literature. In particular, Shalev-Shwartz, Shamir, Srebro, and Sridharan (2009) showed a construction where there is a unique Empirical Risk Minimizer (ERM) that doesn't generalize. Since the ERM is unique and overfitting, *all* or *generic* ERMs fail. However, there are two caveats in that construction. First, it is provided in dimension that is exponential in the sample size. Indeed, Feldman (2016) asked, and left as an open problem, whether such a construction exists in dimension that is linear in the sample size. Second, if we relax the problem a little bit, and allow *approximate* ERMs, even with negligibly small error (exponentially small), already the answer becomes unclear, as some approximate ERMs learn and even many so-called generic solutions (in fact even $w = 0$ in some versions of the construction) generalize well.

In this paper we revisit the problem and we construct a convex learning instance, in linear dimension, where there exists a unique minimizer of the empirical risk that has large generalization error. This resolves Feldman's open problem. But we also extend this and show that even if we take an approximate ERM, with error of order $O(1/(m\sqrt{m}))$ where $m$ is the sample size, it still *must* overfit.

Thus, we obtain that a generic ERM, even approximate-ERM are not a good fit. One criticism that arises is that we look at *generic* ERMs, but not on *generic* problems. In particular, our result shows that there *exists* a learning instance where all ERMs fail, but clearly doesn't tell that on every convex problem this happens. Our focus on worst-case SCO problems stems from the Algorithms we study. Algorithms such as SGD, regularized ERM (Bousquet & Elisseeff, 2002; Shalev-Shwartz et al., 2009), or stable versions of GD (Bassily et al., 2020) all converge to generalizing solutions on *every* convex learning problem with sample size that is independent of the dimension. As such, the question is whether convex loss landscapes have some *shared* property that enables these algorithms to learn. Our study shows that their success is independent from some *generic* property of the empirical minimizer, and succeed to learn *even when* the Guess & Check algorithm fails in SCO.

**Insights on Gradient Descent**  Beyond Guess & Check, our proof and techniques provide new insights and new lower bounds for existing, first order, learning methods. In more detail, our result shows that any algorithm with extremely small empirical training loss would overfit. That includes algorithms such as Gradient Descent when trained for a long time-horizon. Interestingly, the best known upper bound for constrained Gradient Descent w.r.t training time and sample is $O(\eta\sqrt{T} + \eta T/m)$ due to Bassily, Feldman,

Guzmán, and Talwar (2020). Our lower bound demonstrates that when $\eta T = \Theta(m\sqrt{m})$, GD overfits because of its small training error.

We further develop this, and we use our construction and proof technique to deepen this result. We provide, then, a new lower bound, and we show that the generalization error of GD is lower bounded by $\Omega\left(\sqrt{\eta T/m^{1.5}}\right)$, throughout the curve. This is the first, polynomial, lower bound for constrained Gradient Descent that depicts the overfitting w.r.t training accuracy and sample size. It leaves as an open question to further narrow down the gap between the best known upper bound and lower bound.

## 1.1. Related Work

The study of generalization in overparametrized setup is an active, highly intense field of study often focused on deep learning (Neyshabur et al., 2014; Zhang et al., 2021; Arora et al., 2019; Adlam & Pennington, 2020; Terjék & González-Sánchez, 2022).

Early works by Shalev-Shwartz et al. (2009); Feldman (2016) studied generalization in SCO and showed that learnability can be separated from ERMs. Subsequently, SCO has been further investigated, as a benchmark model for overparametrization (Livni, 2023; Koren et al., 2022; Schliserman et al., 2024; Sekhari et al., 2021; Vansover-Hager et al., 2025). Many of these works aim to identify the properties that govern generalization, focusing on implicit bias (Dauber et al., 2020; Amir et al., 2022; Sekhari et al., 2021), the role of regularization (Amir et al., 2021b; Shalev-Shwartz et al., 2009; Livni, 2024), worst-case ERM bounds (Carmon et al., 2024) or other important properties such as stability or information-theoretic bounds (Schliserman et al., 2024; Amir et al., 2021a; Livni, 2024; Attias et al., 2024; Vansover-Hager et al., 2025).

Our work takes a similar vein and looks at SCO as a case study to understand the behavior of *typical* ERMs. Outside the context of convexity, studying random interpolating networks have been a central theme in both empirical and in several theoretical case studies (Chiang et al., 2022; Valle-Perez et al., 2018; Mingard et al., 2021; Harel et al., 2024; Buzaglo et al., 2024; Feder et al., 2025).

**The Sample Complexity of ERMs**  A line of work studies the sample complexity of empirical risk minimizers (ERMs). Shalev-Shwartz et al. (2009) first showed that ERMs may fail to generalize even with sufficient samples to learn. Feldman (2016) later demonstrated that this phenomenon can occur in dimension linear in the sample size. That work also suggested a separation between uniform convergence and ERMs sample complexity. Later Carmon, Yehudayoff, and Livni (2024) provided a first upper bound that confirmed

this separation and tightly characterized the sample complexity of ERMs. Subsequently, Livni (2024) showed that this worst-case ERM behavior can be realized by natural algorithms, proving that vanilla GD generalizes no better than worst-case ERM.

All these results construct instances where some ERMs fail while others succeed. In contrast, our construction focuses on the generalization of the *best-case* ERM. The main exception is Shalev-Shwartz et al. (2009), which presents an exponential-dimensional instance with a unique ERM that fails to generalize. However, in that setting, even an approximate ERM with exponentially small error does generalize.

**Generalization Bounds for Gradient Descent**  Our final application is a new generalization lower bound for Gradient Descent. The sample complexity of Gradient Descent, particularly on non-smooth problems, was studied by Bassily et al. (2020) that gave the first upper bound. Later, its sample complexity was studied in Amir et al. (2021a;b) that showed that in exponential dimension it can overfit. Livni (2024) gave a generalization lower bound in linear dimension, but a gap of $O(\eta T/m)$ remained between the best known upper bound and the lower bound. Amir et al. (2021a) observed that, in exponential dimension, when the training is exponentially long GD will overfit. Here we close the gap further and show a generalization lower bound of order $\Omega(\sqrt{\eta T/m^{1.5}})$. Importantly, we consider *constrained* (i.e. projected) GD. As Zhang et al. (2023); Nikolakakis et al. (2025) observe, when GD is applied without projections, even in 1 dimension (often considered an *underparameterized setting*), GD will have generalization error of $O(\eta T/m)$. In this example, overfitting occurs once the solution norm reaches $\Omega(\sqrt{m})$. Overall then, in this setup, the loss values may become arbitrarily large in which case there is no hope for learnability. As such, this example exposes how learnability becomes ill-posed in unconstrained optimization, rather than revealing a genuine algorithmic limitation.

**Conflict of Interest Disclosure**  The authors declare no financial conflicts of interest.

## 2. Preliminaries and Setup

We consider the standard Stochastic Convex Optimization (SCO) framework (Shalev-Shwartz et al., 2009), with an arbitrary finite[1] instance space $\mathcal{Z}$ and a parameter set

$$\mathcal{W}_d := \{w \in \mathbb{R}^d : \|w\| \le 1\},$$

defined in $d$-dimensional Euclidean space. We assume that there exists a loss function $f : \mathcal{W}_d \times \mathcal{Z} \to \mathbb{R}$ that is convex and $L$-Lipschitz in its first argument. Recall that a function

---

[1]since we are concerned with a lower bound, the finiteness is without loss of generality here

$f$ is convex in $\mathcal{W}_d$ if for every $w_1 \in \mathcal{W}_d$, there exists $g \in \mathbb{R}^d$ such that:

$$\forall w_2 \in \mathcal{W}_d, \quad f(w_1) \le f(w_2) + \langle g, w_1 - w_2 \rangle.$$

The set of $g$'s that satisfy the above equation at $w_1$ is called the set of subdifferentials at $w_1$ and is denoted $\partial f(w_1)$. If $\partial f(w_1)$ contains a single vector then $f$ is differentiable at $w_1$ and the subdifferential is the derivative and is denoted $\nabla f(w_1)$. Finally, $f$ is called $L$-Lipschitz if every subgradient $g$ satisfies $\|g\| \le L$. Equivalently

$$\forall w_1, w_2, \quad |f(w_1) - f(w_2)| \le L\|w_1 - w_2\|.$$

In addition, we say that the function $f$ is $\lambda$-strongly convex if for all $w_1 \in \mathcal{W}_d$ there exists $g \in \partial f(w_1)$ such that for any $w_2 \in \mathcal{W}_d$,

$$f(w_1) \le f(w_2) + \langle g, w_1 - w_2 \rangle - \frac{\lambda}{2}\|w_2 - w_1\|^2, \quad (1)$$

**Learning.**  Our focus is on *learning algorithms*, where a learning algorithm is described as an algorithm $\mathcal{A}$ that receives a finite input sample $S = (z_1, \ldots, z_m) \in \mathcal{Z}^m$, and outputs a parameter $w_S \in \mathcal{W}_d$. Our key assumption is that the sample is drawn i.i.d. from some unknown distribution $D$, i.e. $S \sim D^m$ and the goal of the learner is to minimize the population loss

$$F(w) := \mathbb{E}_{z \sim D}[f(w, z)]. \quad (2)$$

We will say that a learner has sample complexity $m(\varepsilon)$ if, for any $m \ge m(\varepsilon)$, whenever $|S| \ge m$, with probability at least $1/2$:

$$F(w_S) - \min_{w \in \mathcal{W}_d} F(w) \le \varepsilon \quad (3)$$

**Empirical Risk Minimizers.**  A standard approach in learning is to use an *Empirical Risk Minimizer* (ERM). Given a sample $S = (z_1, \ldots, z_m) \in \mathcal{Z}^m$, the empirical risk is defined as

$$F_S(w) := \frac{1}{m} \sum_{i=1}^m f(w, z_i). \quad (4)$$

Given a sample $S$, a parameter $w_S \in \mathcal{W}_d$ is an $\varepsilon$-ERM solution if it satisfies:

$$F_S(w_S) \le \min_{w \in \mathcal{W}_d} F_S(w) + \varepsilon.$$

A $0$-ERM solution is simply called an ERM solution. In general, an $\varepsilon$-ERM solution need not generalize to the population loss. It is known that there are ERMs that fail to generalize without dimension-dependent sample complexity. In particular, the min max rate generalization for ERMs is known to be (Feldman, 2016; Carmon et al., 2024):

$$\mathbb{E}_{S \sim D^m}\left[F(\hat{w}_S) - \min_{w \in \mathcal{W}_d} F(w)\right] = \tilde{\Theta}\left(\frac{d}{m} + \frac{1}{\sqrt{m}}\right). \quad (5)$$

Specifically, Feldman (2016) showed that there exists a learning instance such that, unless $\Omega(d)$ examples are observed, there exists w.h.p. a solution for the empirical risk that fails to generalize.

**Non-ERM learners** In many learning setups such as PAC learning (Vapnik & Chervonenkis, 1971; Valiant, 1984) ERMs are known to be (up to logarithmic factors) min-max optimal. Namely, the statistical rate of learning is the same as the statistical rate of the worst-case ERMs. In SCO, though, this is not the case.

One method that improves over ERM is *regularized ERM*. Bousquet & Elisseeff (2002) showed that when $\hat{f}(\cdot, z)$ is $\lambda$-strongly convex then an (exact) ERM solution satisfies

$$\mathop{\mathbb{E}}_{S \sim D^m}\left[\hat{F}(\hat{w}_S) - \min_{w \in \mathcal{W}_d} \hat{F}(w)\right] = O\left(\frac{1}{\lambda m}\right). \quad (6)$$

As such, we can consider the minimizer of non-strongly convex objective with added regularization term:

$$\min_{w \in \mathcal{W}_d} \frac{\lambda}{2}\|w\|^2 + F_S(w).$$

Indeed, (Bousquet & Elisseeff, 2002) observed that, because the minimizer of the regularized objective is an $O(\lambda)$-ERM for the unregularized objective, if we choose $\lambda = O(1/\sqrt{m})$ we obtain a solution $w_S^\lambda$ such that:

$$\mathbb{E}\left[F(w_S^\lambda) - \min_{w \in \mathcal{W}_d} F(w)\right] = O\left(\frac{1}{\sqrt{m}}\right),$$

This significantly improves over the $\Omega(d/m)$-sample complexity of ERMs in the overparametrized setting.

Regularized ERM is not an ERM algorithm but an approximate ERM. We mention, then, that there are even natural learning algorithms (in fact SGD) that are inherently not ERMs, not even approximate ERMs (Vansover-Hager et al., 2025), that obtain the optimal minmax statistical rate (and are in fact more efficient in some sense (Amir et al., 2021b)).

**Gradient Descent.** Another important optimization algorithm in this context is Gradient Descent (GD) which can also constitute an $\varepsilon$-ERM. GD is a *first-order* algorithm, which means that given a point $w \in \mathcal{W}_d$ and a sample $z \in \mathcal{Z}$ we assume *first-order oracle access* to the function $f(w, z)$. In a nutshell, it means that we can query a (sub)-gradient $g \in \partial f(w, z)$, and in particular, we can calculate a subgradient $g \in \partial F_S(w)$. We refer the reader for further details and background to any standard textbook in the field (Hazan, 2016; Bubeck, 2015; Nemirovsky & Yudin, 1983).

The algorithm operates on the empirical risk and is parameterized by a step size $\eta > 0$ and a number of iterations $T \in \mathbb{N}$. In more detail, given a sample $S = (z_1, \ldots, z_m)$,

GD initializes at $w_0 = 0$ and performs $T$ updates: at step $t \geq 1$ it queries $g_t \in \partial F_S(w_{t-1})$ and performs

$$w_t = \Pi(w_{t-1} - \eta g_t). \quad (7)$$

where $\Pi$ denotes projection onto the unit ball, and finally, the algorithm outputs

$$w_S^{\text{GD}} = \frac{1}{T} \sum_{t=1}^{T} w_t.$$

We note in passing that our results apply to any suffix-averaged output such as, for example, last iterate.

With correct choice of hyperparameters, GD is a case of an $\varepsilon$-ERM. In particular, a classical result in convex optimization (Nemirovsky & Yudin, 1983) yields an optimization guarantee of

$$F_S(w_S^{\text{GD}}) - \min_{w \in \mathcal{W}_d} F_S(w) = \Theta\left(\eta + \frac{1}{\eta T}\right). \quad (8)$$

Notice that a choice of $\eta = 1/\sqrt{T}$ and $T = 1/\varepsilon^2$ provides an $O(\varepsilon)$-ERM algorithm.

The generalization performance of Gradient Descent has also been studied. In particular, for general convex $L$-Lipschitz losses, Bassily et al. (2020) analyzed GD via algorithmic stability and proved that

$$\mathop{\mathbb{E}}_{S \sim D^m}\left[F_S(w_S^{\text{GD}}) - F(w_S^{\text{GD}})\right] = O\left(\eta\sqrt{T} + \frac{\eta T}{m}\right). \quad (9)$$

Observe that Equations (8) and (9) suggest a learning algorithm. Specifically, by choosing $T = m^2$ and $\eta = 1/(m\sqrt{m})$, we obtain a stable version of GD that minimizes the population error with error $O(1/\sqrt{m})$, given $m$ samples. Again, similar to regularized ERM, it is a $O(1/\sqrt{m})$-ERM and not an ERM. Despite these similarities, in important technical respects, GD differs fundamentally from a regularized or implicitly biased algorithm (Dauber et al., 2020).

Subsequent lower bounds (Amir et al., 2021b; Livni, 2024) showed that the $\eta\sqrt{T}$ term is unavoidable, establishing its tightness. In particular, taking both optimization and generalization into account, (Livni, 2024) demonstrated that there exists a learning instance where

$$\mathop{\mathbb{E}}_{S \sim D^m}\left[F(w_S^{\text{GD}}) - \min_{w \in \mathcal{W}_d} F(w)\right] = \Omega\left(\eta\sqrt{T} + \frac{1}{\eta T}\right). \quad (10)$$

Notice that Equation (10) does not rule out taking $T$ to infinity if $\eta < 1/\sqrt{T}$. On the other hand, Equation (9) hints that as $T$ goes to infinity generalization should deteriorate. For very large $T = \Omega(2^m)$, Amir et al. (2021b)'s result showed that GD may overfit and provide low empirical risk but high generalization error. However, this result leaves a significant gap from the upper bound and does not rule out generalization even for $T = \Omega(m)$.

## 3. Main Results

We next present our main results, which provide complementary lower bounds to Equation (6) and Equation (9).

### 3.1. Generalization Lower Bounds for Empirical Risk Minimizers

**Theorem 3.1.** *Fix any $m \in \mathbb{N}$, then there exists $\varepsilon = \Theta(m^{-3/2})$ and a finite instance space $\mathcal{Z}$, a distribution $D$ over $\mathcal{Z}$, and a 1-Lipschitz loss $f : \mathcal{W}_d \times \mathcal{Z} \to \mathbb{R}$ in dimension $d = 6 \cdot m$ such that, with probability at least $1/2$ over $S \sim D^m$, every $\varepsilon$-ERM solution $w_S$ satisfies*

$$F(w_S) - \min_{w \in \mathcal{W}_d} F(w) \geq \Omega(1).$$

*Moreover, $f$ is $\lambda$-strongly convex with $\lambda = m^{-3/2}$.*

This resolves an open question by Feldman (2016, Remark 3.4). Indeed, because any exact ERM is also an $\varepsilon$-ERM, the result yields an instance where the (exact) ERM is *unique*, due to strong convexity, and incurs a constant excess-risk gap. But more generally, every $\Theta(m^{-3/2})$-ERM solution incurs a constant population excess risk and therefore fails to generalize. This is the first results that applies to approximate ERMs with accuracy that is inverse polynomial in sample size.

Notice that by Equation (6) the generalization error of an ERM over a $\lambda$-strongly convex function is given by $O(1/\lambda m)$. A natural question, is then to obtain refined lower bounds that depend on the curvature. The following theorem provides such refinement:

**Theorem 3.2.** *Fix any $m \in \mathbb{N}$ and let $m^{-3/2} \leq \lambda \leq m^{-1/2}$, then there exists $\varepsilon = \Theta\left(\frac{1}{\lambda m^3}\right)$ and a finite instance space $\mathcal{Z}$, a distribution $D$ over $\mathcal{Z}$, and a $\lambda$-strongly convex, 1-Lipschitz loss $f : \mathcal{W}_d \times \mathcal{Z} \to \mathbb{R}$ in dimension $d = 6 \cdot m$ such that, with probability at least $1/2$ over $S \sim D^m$, any $\varepsilon$-ERM solution $w_S$ satisfies*

$$F(w_S) - \min_{w \in \mathcal{W}_d} F(w) \geq \Omega\left(\frac{1}{\lambda m^{3/2}}\right).$$

Distinctively from Theorem 3.1, Theorem 3.2 does not demonstrate a constant generalization error, instead it provides a fine-grained suboptimality bound for the ERM solution. For example, fix $\lambda = m^{-1.01}$, our result demonstrates a suboptimal statistical rate for $O(m^{-1.99})$-ERMs of $\Omega(m^{-0.49})$.

The condition $\lambda \geq m^{-3/2}$ is not restrictive. For $\lambda \leq m^{-3/2}$, Theorem 3.1 shows that there exists a $\lambda$-strongly convex (in fact $m^{-3/2}$-strongly convex) where the population excess risk is already a constant, which is the worst-case bound.

### 3.2. Applications and Extensions to Gradient Descent

Since Theorem 3.1 applies to *any* $\varepsilon$-ERM, it also applies to Gradient Descent if it achieves $\varepsilon$-accuracy. We summarize this in the following corollary, which gives a new lower bound for GD and exponentially narrows the gap between the upper bound in Equation (9) and the best previously known lower bounds (Amir et al., 2021a).

**Corollary 3.3.** *Fix any $m, T \in \mathbb{N}$, and $\eta > 0$ such that $\eta T = \Omega(m^{3/2})$, then there exist a finite instance space $\mathcal{Z}$, a distribution $D$ over $\mathcal{Z}$ and a 1-Lipschitz convex loss $f : \mathcal{W}_d \times \mathcal{Z} \to \mathbb{R}$ defined in $d = \Omega(m + T^{1/3})$, such that if we run GD for $T$ steps, with learning rate $\eta$, then with probability at least $\frac{1}{2}$ over $S \sim D^m$,*

$$F(w_S^{\mathrm{GD}}) - \min_{w \in \mathcal{W}_d} F(w) = \Omega(1).$$

To see Corollary 3.3 follows from Theorem 3.1, recall from Equation (8) that GD has an optimization error:

$$F_S(w_S^{\mathrm{GD}}) - \min_{w \in \mathcal{W}_d} F_S(w) = \Theta\left(\eta + \frac{1}{\eta T}\right).$$

It is also known (Livni, 2024) that the generalization error of GD is bounded by $\Omega(\eta\sqrt{T})$. As such, we can assume without loss of generality that $\eta < 1/\eta T$. In particular, the optimization error of GD is dominated by the term $O(\frac{1}{\eta T})$. Thus, if $\eta T \geq \Omega(m^{3/2})$ we conclude that GD is an $O(m^{-3/2})$-ERM and the conclusion follows from Theorem 3.1. $\qquad\square$

While the above corollary yields a lower bound in the regime $\eta T = \Omega(m^{3/2})$, it leaves open the intermediate horizons. In particular $\eta T < m^{3/2}$. We address this remaining regime via a direct analysis that also improves the dependence on the dimension:

**Theorem 3.4.** *Fix any $m, T \in \mathbb{N}$, $\eta \in (0, 1)$, satisfying $\eta T > \sqrt{m}$. Then for $d = 6 \cdot m$ there exist a finite instance space $\mathcal{Z}$, a distribution $D$ over $\mathcal{Z}$ and a convex loss $f : \mathcal{W}_d \times \mathcal{Z} \to \mathbb{R}$ that is 1-Lipschitz in $w \in \mathcal{W}_d$, such that for $S \sim D^m$, if we run GD for $T$ steps, with step size $\eta$, then with probability at least $\frac{1}{2}$ over $S$,*

$$F(w_S^{\mathrm{GD}}) - \min_{w \in \mathcal{W}_d} F(w) = \Omega\left(\min\left\{\sqrt{\frac{\eta T}{m^{3/2}}}, 1\right\}\right).$$

Together with the work of (Livni, 2024), we obtain a generalization lower bound for GD of $\Omega\left(\eta\sqrt{T} + \sqrt{\frac{\eta T}{m^{3/2}}}\right)$ that is complementary to the best known upper bound in Equation (9). We leave it as an open problem to further close the gap.

When $\eta T < \sqrt{m}$, notice that Theorem 3.4 cannot hold. In particular we obtain that $\sqrt{\frac{\eta T}{m^{3/2}}} \geq \frac{\eta T}{m}$ which contradicts

(Bassily et al., 2020). However in this regime, by Equations (9) and (10), since $\eta T/m < 1/\eta T$, we already have tight characterization of the sample complexity:

$$\mathbb{E}_{S \sim D^m} \left[ F(w_S^{\mathrm{GD}}) - \min_{w \in \mathcal{W}_d} F(w) \right] = \Theta \left( \eta \sqrt{T} + \frac{1}{\eta T} \right). \tag{11}$$

## 4. Discussion

This work explores the contribution of the loss landscape to generalization in Stochastic Convex Optimization. While at first sight a convex loss landscape sounds like an exemplary case of a well-behaved loss landscape, our investigation shows a much more intricate story. Indeed, convexity enables learning, and generalization happens at a much faster rate than expected from standard uniform convergence argument or capacity control. Nevertheless, to learn, we need to craft specially designed algorithms, as generic properties of the empirical risk minimizing solutions do not guarantee learning. The underlying question, then, remains: what enables learning of stochastically convex problems. It turns out that algorithmic properties such as implicit bias (Dauber et al., 2020; Vansover-Hager et al., 2025), regularization (Amir et al., 2021b;a) or information-theoretic bounds (Livni, 2023; Attias et al., 2024) also fall short to explain generalization in SCO. Overall, our result leave open several questions, as well as future directions of research that we next describe.

**Approximate ERMs** We showed a construction where all $\varepsilon$-ERM solutions with $\varepsilon = O(m^{-3/2})$ fail to generalize. In any problem, there is always an $O(m^{-1/2})$-ERM solution that generalizes (e.g regularized-ERM and stable-GD). This leaves an open question whether for $m^{-3/2} < \varepsilon < m^{-1/2}$ there exists a problem where every $\varepsilon$-ERM solution fails.

**Strongly Convex Objectives** We provided a new generalization lower bound for strongly convex problems. Theorems 3.1 and 3.2, together, show that When $\lambda \leq 1/\sqrt{m}$, the ERM will have generalization error of $\Omega(\frac{1}{\lambda m^{3/2}})$. One can show that when $\lambda \geq 1/\sqrt{m}$, the sample complexity of the ERM solution is $\Theta\left(\frac{1}{\lambda m}\right)$.

But for $\lambda \leq 1/\sqrt{m}$, comparing with Equation (6), it remains open what is the tight bound. Specifically, if there exists a strongly convex construction such that the generalization error of the ERM is at least $\Omega(\frac{1}{\lambda m})$.

**Smooth Objectives** Our result closes a gap, and provides a first polynomial lower bound for long training. A natural open problem is to further close this gap. Moreover, for smooth objectives, it is known (Hardt et al., 2016) that gradient descent enjoys an improved generalization bound of $O(\eta T/m)$, improving over $\Theta(\eta \sqrt{T} + \eta T/m)$ (Bassily et al., 2020). Consequently, extending our results to the smooth setting is an important challenge. At present, our lower bounds do not apply to smooth objectives.

## 5. Technical Overview

We next outline our main construction and proof techniques. The full proofs are provided in Sections A and B. Here we focus on providing the necessary technical background and intuition about the construction. Therefore, after providing necessary definitions, we outline a proof for Theorem 3.1 in the special case of an exact ERM, i.e. $\varepsilon = 0$. This is perhaps the most illustrative and interesting instance of the result. Also, to simplify the exposition, we will provide the construction in dimension $d = O(m^2)$ and not in dimension that is linear in $m$. These simplifications are meant to help avoid small technical details and highlight the main ideas in the argument.

### 5.1. Feldman's function

Our construction shows that *any* ERM fails. Therefore, as a first step, we require a construction that demonstrate that *some* ERMs fail. This construction was provided by Feldman, and it is the first building block in our lower bound, and we begin this overview by a brief outline of the construction with the necessary adaptations for our setting:

**Asymptotically good binary codes.** In general, Feldman's functions are defined by a code and they return the correlation between the input and a most correlated codeword within a random subset of codewords. The simplest construction uses a random code and random subsets. However, for our purposes, we need to work with a more carefully constructed family of codeword subsets. To this end, it is convenient to consider a full coding scheme, rather than just individual codewords. We therefore follow Feldman, who use *asymptotically good codes*, which can also be computed efficiently. In more detail, for $k \in \mathbb{N}$, an asymptotically good code with relative distance $\rho$ is a mapping:

$$G_k : \{-1, 1\}^k \longrightarrow \{-1, 1\}^{2k} \tag{12}$$

such that for any two distinct points in the range (referred to as codeword): $G_k(u), G_k(v) \in \{-1, 1\}^{2k}$, their Hamming distance is at least $\rho \cdot 2k$. In particular,

$$\langle \overline{G_k(u)}, \overline{G_k(v)} \rangle < 1 - \frac{\rho}{2},$$

where we denote $\overline{x} = x/\|x\|$. It is known Justesen (1972); Sipser & Spielman (1996) that for some *constant* $0 < \rho < 1/2$ an asymptotically good code exists for every $k$. Moreover, $G_k$ can be computed efficiently. Throughout, we denote by $\rho$ the existing universal constant, and when $k$ is fixed and there is no room for confusion, we will denote by

$G$ an $\rho$-asymptotically good code, and neglect the subscript $k$.

**Feldman's function.** Next, given a fixed asymptotically good code, we describe Feldman's function. For each coordinate $i \in [k] := \{1, ..., k\}$, define the set

$$W_i := \{G(v) \; : \; v \in \{-1,1\}^k, \; v(i) = 1\}. \quad (13)$$

For any $\zeta \in (0, 1]$, we define the function: $h^\zeta : \mathcal{W}_{2k} \times [k] \to \mathbb{R}$ by

$$h^\zeta(w, i) := \max\left\{\zeta\left(1 - \frac{\rho}{2}\right), \max_{x \in W_i}\langle w, \overline{x}\rangle\right\}. \quad (14)$$

where $\rho$ is the universal constant defined above. By construction, for every $i \in [k]$, the function $h^\zeta(\cdot, i)$ is convex and 1-Lipschitz over the Euclidean unit ball $\mathcal{W}_{2k}$. Given a distribution $D$ over $[k]$, we denote the corresponding population objective by

$$h_D^\zeta(w) := \mathbb{E}_{z \sim D}[h^\zeta(w, z)],$$

and similarly, given a sample $S = (z_1, \ldots, z_m) \sim D^m$, we write for the empirical distribution:

$$h_S^\zeta(w) := \frac{1}{m}\sum_{i=1}^m h^\zeta(w, z_i)$$

Notice that, given a random uniform finite sample $z_1, \ldots, z_m \in [k]^m$, when $m \leq k/2$, there exists $G(v) \notin \bigcup_{j=1}^m W_{z_j}$ such that for every $i \notin \{z_1, \ldots, z_m\}$, $G(v) \in W_i$. In particular, set $\zeta = 1$, by the property of the code, we have that $h_S^\zeta(\overline{G(v)}) \leq (1 - \rho/2)$, but

$$h_D^\zeta(\overline{G(v)}) \geq \frac{1}{2}\left(1 - \frac{\rho}{2}\right) + \frac{1}{2} \geq h_S^\zeta(\overline{G(v)}) + \frac{\rho}{4}.$$

As such, with appropriate choice of $\zeta$, Feldman's function demonstrates the existence of an ERM that fails to generalize. However, also notice that $h_D^\zeta(0) = h_S^\zeta(0) = \min_w h_S^\zeta(w)$ is a valid ERM that *does* generalize.

## 5.2. A simple proof for exact ERMs in quadratic dimension

As discussed, we next outline the construction and the main proof ideas behind Theorem 3.1.

To simplify the presentation, we focus, only in the presentation, on the case of an exact ERM, and we prove the result for $d = \Omega(m^2)$ which simplifies some of the calculations. The full proof is deferred to Section A.

Before we delve into the proof let us briefly outline the challenges. As discussed, Feldman's function already has bad ERMs, however it also has *good* ERMs, in particular, 0

is a good ERM. One idea to mitigate this, is to add a noisy linear term to the function. Consider then a function of the form

$$h^\zeta(w, i) + \langle w, \delta_i\rangle,$$

where $\delta_i$ is some noise. This linear term already guarantee that 0 is no longer the minimizer of any random empirical sample (recall that we now consider *exact* ERMs). Further, if by some *luckiness* the empirical noise vector $\frac{1}{m}\sum \delta_z$ is sufficiently correlated with the *bad* ERM, which we will denote by $b_S$, then we obtain a bad unique minimizer on the sphere. Unfortunately though, such strategy cannot work. No matter how the distribution over $\delta_i$ is defined, the correlation between the empirical term $\frac{1}{m}\sum \delta_z$ and each potentially bad ERM $b_S$ is concentrated around its *true* correlation *uniformly*. This follows from a standard uniform convergence argument for linear losses. Now, we cannot have a distribution where the noise is correlated with all possible $b_S$, and because $b_S$ depends on the sample, we cannot apriori choose the noise to be correlated with the bad ERM in hindsight.

As such, we take a different approach. We still use a noise vector that depends on the sample. Then, we use a *link* function that identifies the sample from the noise, and translates it to a bad ERM. This link function is the crux of our proof, and the fact that we can build such a convex link function is what enables the construction. In more details, our construction for now resides in dimension $d = 6m^2$ (again, in the full proof we assume $d = O(m)$), and we decompose the parameter as

$$w = (w^c, w^m) \in \mathbb{R}^{2k} \times \mathbb{R}^k, \quad k = 2m^2$$

$w^m$ can be thought of as a *message* term and it will encode the sample via noise. Then, $w^c$ can be thought of as a *code* term that will contain a bad ERM for Feldman's function at the minimum. For every $i \in [k]$, set:

$$e_i(j) = \begin{cases} 0 & j \neq i \\ 1 & j = i \end{cases}, \qquad v_S = \sum_{z \in S} e_z,$$

$$v_S^s(i) = \begin{cases} -1 & i \in S \\ +1 & i \notin S \end{cases}.$$

And, we assume $i$ are chosen randomly and uniformly. We let $\zeta = 1/2$, and we consider the following function, and the corresponding empirical risk:

$$f(w, i) = h^\zeta(w^c, i) - \langle w^m, e_i\rangle + \max\{p(w), 0\}$$
$$+ \frac{1}{2\sqrt{m}}\|w^m\|^2 + \gamma\|w^c\|^2$$

$$F_S(w) = h_S^\zeta(w^c) - \frac{1}{m}\langle w^m, v_S\rangle + \max\{p(w), 0\}$$
$$+ \frac{1}{2\sqrt{m}}\|w^m\|^2 + \gamma\|w^c\|^2$$

where

$$p(w) = \max_{\{S' \in \mathcal{S}\}} \left\{ \frac{1}{2m} \langle v_{S'}, w^m \rangle - \gamma \langle \overline{G(v_{S'}^s)}, w^c \rangle \right\},$$

and $\mathcal{S}$ is the set of all samples that don't contain collisions, i.e. $S \in \mathcal{S}$ if for every $z_i, z_j \in S$, $z_i \neq z_j$. Now, throughout our analysis, we will assume that there are no collisions in the instance sample $S$, i.e. $S \in \mathcal{S}$. This is where we use the fact that $d = O(m^2)$, and we can assume this event happens with constant probability due to (lack of) birthday paradox. It helps simplify the analysis because we can assume that

$$\|v_S\| = \sqrt{m}.$$

Notice, because of the linear term, $w^m$ is incentivized to be correlated with $v_S$. The function $p$, though, when the input is *most* correlated with $v_S$, penalizes such correlation but also incentivizes correlation with $\overline{G(v_S^s)}$ in the code term. This is what we will need for overfitting. But to reason that the most correlated vector is $v_S$ we will need the first term (i.e. the message term) in $p$ to be the dominant term. In particular, assume the parameter $\gamma$ to be sufficiently small. Smaller than any potential loss we might encounter when choosing $S' \neq S$. I.e.

$$\gamma < \min_{S' \neq S} \left\{ \frac{1}{4m\sqrt{m}} \left( \|v_S\|^2 - \langle v_S, v_{S'} \rangle \right) \right\}. \tag{15}$$

That $\gamma$ can be chosen to be positive follows from Cauchy Schwartz inequality, and finiteness of the possible samples. Importantly, we only care in this proof overview for the exact ERM and we don't care about the strong convexity. When these factors come into play, we will need to choose $\gamma$ a little bit more carefully, to ensure large strong convexity and extension to approximate ERMs. Now, let us guess that at the minimizer $w^\star$, we first have that $\nabla h_S((w^\star)^c) = 0$ and that $v_S$ is indeed the maximizer of the $p$ term and, in particular:

$$\nabla p(w^\star) = \left( -\gamma \overline{G(v_S^s)}, \frac{1}{2m} v_S \right).$$

Deriving $F_S$, equating to zero, we obtain a closed form solution of our candidate $w^\star$:

$$0 = -\frac{1}{m} v_S + \frac{1}{2m} v_S + \frac{1}{\sqrt{m}} w^m \Rightarrow (w^\star)^m = \frac{1}{2\sqrt{m}} v_S.$$

And,

$$0 = -\gamma \overline{G(v_S^s)} + 2\gamma w^c \Rightarrow (w^\star)^c = \frac{1}{2} \overline{G(v_S^s)}.$$

To conclude that our candidate is indeed the minimizer, we need to verify that our assumptions indeed hold. If they hold, then by first order optimality conditions $w^\star$ is indeed the minimizer. Indeed, we have that $\nabla h_S^\zeta(\frac{1}{2} \overline{G(v_S^s)}) = 0$, by our

choice $\zeta = 1/2$, the fact that $v_S^s$ is such that $G(v_S^s) \notin W_{z_i}$ for any $z_i \in S$ and the properties of asymptotically good code.

Next, to see that $v_S$ is indeed the maximizer, by Cauchy Schwartz for every $S' \neq S$:

$$\frac{1}{2m} \langle v_{S'}, (w^\star)^m \rangle - \gamma \langle \overline{G(v_{S'}^s)}, (w^\star)^c \rangle$$
$$\leq \frac{1}{4m^{1.5}} \langle v_{S'}, v_S \rangle + \frac{\gamma}{2}$$
$$< \frac{1}{4m^{1.5}} \langle v_S, v_S \rangle - \frac{\gamma}{2} = p(w^\star),$$

Where the second inequality is due to Equation (15) and writing $\frac{\gamma}{2} = \gamma - \frac{\gamma}{2}$.

### 5.3. Gradient Descent

We now illustrate how the techniques we developed can be further extended to provide generalization bounds for Gradient Descent. Again, we will simplify here and provide a construction in $d = m^2$. Moreover, we will only illustrate how our argument works when $\eta T$ diverges, without the achieving the explicit $\Omega(\sqrt{\eta T / m \sqrt{m}})$ bound stated in Theorem 3.4.

Notice that the construction in Section 5.2, being strongly convex, already establishes such a bound as $T$ diverges. So, we only focus on showing how our argument can be applied to the trajectory, as this forms the basis of our final proof presented in Section B.

We work with the same notations as before, and our function takes a slightly different form:

$$f(w, i) = h^\zeta(w^c, i) - \langle w^m, e_i \rangle + \max\{p(w), 0\} + \gamma^c \|w^c\|_2^2,$$

where we now don't regularize the message term. The function $p$ also takes a slightly different form:

$$p(w) = \max_{\{S' \in \mathcal{S}\}} \left\{ \gamma^m \langle v_{S'}, w^m \rangle - \gamma^c \langle \overline{G(v_{S'}^s)}, w^c \rangle \right\},$$

where we choose $\gamma^m = \frac{1}{m} - \frac{1}{2\eta T \sqrt{m}}$. The difference in the analysis from before, is that now we consider the iterates of Gradient Descent instead of the minimizer. But we follow a similar pattern. First, observe that $p(w_0) = p(0) = 0$ which entails that $0 \in \partial \max\{p(w_0), 0\}$. Next, we assume that the trajectory follows a rule such that for every $t \geq 1$:

$$\nabla p(w_t) = \left( -\gamma^c \overline{G(v_S^s)}, \gamma^m v_S \right),$$

As before, this amounts to $v_S$ being the maximizer in the expression of $p$. Then, by induction, one can show that for

$t \geq 1$:

$$w_{t+1}^m = w_t^m - \eta \partial_{w^m} F_S(w_t)$$
$$= \left(\frac{\eta}{m} + \frac{t-1}{2T\sqrt{m}}\right) v_S + \frac{\eta}{m} v_S - \eta \gamma^m v_S$$
$$= \left(\frac{\eta}{m} + \frac{t-1}{2T\sqrt{m}}\right) v_S + \frac{1}{2T\sqrt{m}} v_S$$
$$= \left(\frac{\eta}{m} + \frac{t}{2T\sqrt{m}}\right) v_S,$$

and

$$w_{t+1}^c = w_t^c - \eta \partial_{w^c} F_S(w_t)$$
$$= w_t^c + \gamma^c \eta \overline{G(v_S^s)} - 2\gamma^c \eta w_t^c$$
$$= (1 - 2\eta\gamma^c) w_t^c + \gamma^c \eta \overline{G(v_S^s)}$$
$$= (1 - 2\eta\gamma^c) \frac{\left(1 - (1 - 2\eta\gamma^c)^{t-1}\right)}{2} \overline{G(v_S^s)}$$
$$\quad + \gamma^c \eta \overline{G(v_S^s)}$$
$$= \frac{1 - (1 - 2\eta\gamma^c)^t}{2} \overline{G(v_S^s)}$$

Now, again, notice that if we choose $\gamma^c$ sufficiently small. In particular

$$\gamma^c \ll \frac{\eta\gamma^m}{m} \min_{S' \neq S} \left(\|v_S\|^2 - \langle v_S', v_S \rangle\right),$$

then as before the left term of $p$ dominates, and our induction hypothesis holds.

Then, when $T$ is sufficiently large, roughly $T \approx 1/(2\eta\gamma^c)$ then $w_T^c = O(\overline{G(v_S^s)})$ which by the same argument as before, will constitutes a bad minimizer for correct choice of $\zeta$.

## Acknowledgements

This work was supported by the European Union (ERC, FoG 101116258). Views and opinions expressed are however those of the author(s) only and do not necessarily reflect those of the European Union or the European Research Council Executive Agency. Neither the European Union nor the granting authority can be held responsible for them.

## Impact Statement

This work is entirely theoretical and does not involve human subjects, sensitive data, or deployed systems. As such, there are no direct ethical concerns, risks, or societal impacts associated with the research.

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

# A. Proofs of the Empirical Risk Minimizer Lower Bounds

Theorems 3.1 and 3.2 are immediate corollaries of the following technical Lemma.

**Lemma A.1.** *There exists a constant $0 < \rho < 1/2$ (as described in Section 5.1) such that: Fix any $m \in \mathbb{N}$, and let $\lambda \leq \frac{1}{\sqrt{m}}$, then for $d = 6m$ there exist a finite instance space $\mathcal{Z}$, a distribution $D$ over $\mathcal{Z}$ and a $\lambda$-strongly convex loss $f : \mathcal{W}_d \times \mathcal{Z} \to \mathbb{R}$ that is 7-Lipschitz in $w \in \mathcal{W}_d$, such that for $S \sim D^m$, with probability at least $\frac{1}{2}$ over $S$, any $\varepsilon$-ERM solution $w_S$ satisfies*

$$F(w_S) - F(0) \geq \min\left\{\frac{\rho}{72\lambda m^{1.5}}, \frac{\rho}{12}\right\} - 7\sqrt{\frac{2\varepsilon}{\lambda}}.$$

To see that Theorem 3.1 follows, first we can assume w.l.o.g that $m \geq 7$ (otherwise the generalization error is bounded by a standard information-theoretic argument). Apply the Lemma with a choice of $\lambda = \frac{7}{m^{1.5}} \leq \frac{1}{\sqrt{m}}, \varepsilon = \frac{\lambda \rho^2}{4 \cdot 72^2 \cdot 7^4}$. Then we obtain a 7-Lipschitz, $7/m^{1.5}$-strongly convex function such that

$$F(w_S) - F(0) = \Omega(1).$$

Dividing by 7 yields the result.

Similarly, Theorem 3.2 follows by considering $\varepsilon = \frac{\rho^2}{4 \cdot 72^2 \cdot 7^2 \cdot \lambda m^3}$, for every $\frac{7}{m^{1.5}} \leq \lambda < 7/\sqrt{m}$.

As such, we are left with proving Lemma A.1

## A.1. Proof of Lemma A.1

For a fixed $m \in \mathbb{N}$, set $k = 2m$ and $d = 3k \in \Theta(m)$, and let $\lambda \leq \frac{1}{\sqrt{m}}$. We will decompose the parameter vector and denote

$$w = (w^c, w^m) \in \mathbb{R}^{2k+k},$$

where $w^c \in \mathbb{R}^{2k}$ and $w^m \in \mathbb{R}^k$. For our construction, for each $i \in [k]$ we define the following loss function, parameterized by $\zeta, \gamma^c, \lambda^m, \lambda^c \leq 1$, and $\gamma^m \leq \frac{1}{\sqrt{k}}$

$$f(w, i) := h^\zeta(w^c, i) - \langle w^m, \delta_i \rangle + \max\{p(w), 0\} + \frac{\lambda^m}{2}\|w^m\|^2 + \frac{\lambda^c}{2}\|w^c\|^2, \tag{16}$$

where, $h^\zeta(w, i)$ is Feldman's function in $\mathbb{R}^{2k}$, with an asymptotically good code $G = G_k$ as defined in Equation (14):

$$\delta_i(j) = \begin{cases} 1/m - 2 & i = j \\ 1/m & i \neq j \end{cases}, \qquad p(w) := \max_{v \in \{-1,1\}^k}\left\{\gamma^m\langle v, w^m\rangle - \gamma^c\langle\overline{G(v)}, w^c\rangle\right\}.$$

One can observe that each term is convex, moreover $\|\delta_i\| \leq 2$ and $\gamma^m\sqrt{k} \leq 1$, so overall $f$ is 7-Lipschitz in the unit ball, and that $f$ is $\alpha$-strongly convex for $\alpha := \min\{\lambda^m, \lambda^c\}$. Then, we let $D$ be a uniform distribution over $[k]$.

We set for a sample $S$ the vectors:

$$v_S(i) = \sum_{j=1}^m \delta_{z_j}(i) = \begin{cases} 1 - 2|\{j : z_j = i\}| & i \in S \\ 1 & i \notin S \end{cases} \quad ; \quad v_S^s(i) = \begin{cases} -1 & i \in S \\ +1 & i \notin S \end{cases}$$

We will now use the following concentration lemma to bound $\|v_S\|$.

**Lemma A.2** ((Boucheron et al., 2003), example 6.13). *Let $X_1, \ldots, X_m$ be i.i.d random variables with $E[X_i] = \bar{X}$ such that $\|X_i\| \leq C$ w.p. 1, for some constant $C > 0$. Then for all $t > \frac{C}{\sqrt{m}}$:*

$$\mathbb{P}\left(\|\frac{1}{m}\sum_{i=1}^m X_i - \bar{X}\| > t\right) \leq e^{-2\left(\frac{mt^2}{C^2} - 1\right)}$$

Applying Lemma A.2 to the random variable $X_i = \delta_i$ yields that, with probability at least $1/2$,

$$\|v_S\| = \|v_S - m \cdot 0\| = \|v_S - \frac{m}{k}\sum_{i=1}^k \delta_i\| = \|\sum_{i=1}^m \delta_{z_i} - m \cdot \mathbb{E}_{i\sim D}[\delta_i]\| \leq 3\sqrt{m} \tag{17}$$

We throughout assume this event happened, and analyze the generalization error under this event.

**The special case of $\varepsilon = 0$** We begin by proving the statement for the special case of *exact* ERM. The general case then follows easily and is deferred to the end of the proof. Therefore, we now assume that $\varepsilon = 0$ and we only need to show that:

$$F(w_S^\star) - F(0) \geq \min\left\{\frac{\rho}{72m^{1.5}\lambda}, \frac{\rho}{12}\right\}. \tag{18}$$

We start with the following claim

**Claim 1.** *Suppose we choose the parameters of $f(w, i)$ in the following regime:*

$$\frac{9}{\sqrt{m}} \leq \lambda^m < \frac{27}{2\sqrt{m}}, \qquad \gamma^m \leq \frac{1}{2m}, \qquad \gamma^c \leq \frac{\gamma^m}{9\sqrt{m}}, \qquad \lambda^c \geq 3\gamma^c, \qquad \zeta \geq \frac{\gamma^c}{\lambda^c} \tag{19}$$

*Then, the unique minimizer of $F_S$ in $\mathcal{W}_d$ is given by:*

$$(w_S^\star)^c = \frac{\gamma^c}{\lambda^c}\overline{G(v_S^s)} \qquad (w_S^\star)^m = \frac{1}{\lambda^m}\left(\frac{1}{m}v_S - \gamma^m v_S^s\right). \tag{20}$$

Before we prove the claim let us observe how it entails Equation (18). First, observe that for any $w \in \mathcal{W}_d$

$$
\begin{aligned}
F(w) - F(0) &\geq h_D^\zeta(w^c) - h_D^\zeta(0) - \langle w^m, \mathbb{E}[\delta_z]\rangle + \max\{p(w), 0\} \\
&= h_D^\zeta(w^c) - h_D^\zeta(0) + \max\{p(w), 0\} & \mathbb{E}[\delta_z] = 0 \\
&\geq h_D^\zeta(w^c) - h_D^\zeta(0) & 0 \leq \max\{p(w), 0\} \quad (21)
\end{aligned}
$$

Next, use the fact that with probability at least $1 - \frac{m}{k} \geq \frac{1}{2}$, we have that $G(v_S^s) \in W_z$. Indeed, $G(v_S^s) \notin W_z$ only for $\{W_{z_1}, \ldots, W_{z_m}\}$. Therefore, by the definition of Feldman's function:

$$
\begin{aligned}
F(w_S^\star) - F(0) &\geq h_D^\zeta((w_S^\star)^c) - h_D^\zeta(0) & \textit{Equation (21)} \\
&\geq \frac{1}{2}\left(\langle \overline{G(v_S^s)}, (w_S^\star)^c\rangle - \zeta\left(1 - \frac{\rho}{2}\right)\right) & \mathbb{P}(G(v_S^s) \in W_z) \geq 1/2 \\
&= \frac{1}{2}\left(\frac{\gamma^c}{\lambda^c} - \zeta + \frac{\zeta\rho}{2}\right) & (w_S^\star)^c = \frac{\gamma^c}{\lambda^c}\overline{G(v_S^s)}
\end{aligned}
$$

Setting $\zeta = \frac{\gamma^c}{\lambda^c}$ yields

$$\left(\frac{\gamma^c}{\lambda^c} - \zeta + \frac{\zeta\rho}{2}\right) \geq \frac{\rho\gamma^c}{2\lambda^c}$$

and Equation (18) follows by choosing

$$\gamma^m = \frac{1}{2m}, \qquad \gamma^c = \min\left\{\frac{1}{18m^{1.5}}, \frac{\lambda}{3}\right\}, \qquad \lambda^c = \lambda,$$

It remains to show that Equation (20) holds:

**Proof of Claim 1** Averaging (16) over $S$ gives the empirical objective

$$F_S(w) = h_S^\zeta(w^c) - \frac{1}{m}\langle w^m, v_S\rangle + \max\{p(w), 0\} + \frac{\lambda^m}{2}\|w^m\|^2 + \frac{\lambda^c}{2}\|w^c\|^2. \tag{22}$$

Denote

$$\widetilde{w^c} := \frac{\gamma^c}{\lambda^c}\overline{G(v_S^s)}, \qquad \widetilde{w^m} := \frac{1}{\lambda^m}\left(\frac{1}{m}v_S - \gamma^m v_S^s\right). \tag{23}$$

and let us show that $\widetilde{w} = w_S^\star$, where $\widetilde{w} = (\widetilde{w^m}, \widetilde{w^c})$. To begin, we first show that $\widetilde{w}$ is indeed a feasible solution. This follows from a straightforward calculation and the constraint on the parameter set:

$$
\begin{aligned}
\|\widetilde{w}\| &\leq \|\widetilde{w^m}\| + \|\widetilde{w^c}\| \\
&\leq \frac{1}{\lambda^m}\left(\frac{1}{m}\|v_S\| + \gamma^m\|v_S^{\mathrm{s}}\|\right) + \frac{\gamma^c}{\lambda^c}\|\overline{G(v_S^{\mathrm{s}})}\| \\
&\leq \frac{1}{\lambda^m}\left(\frac{3}{\sqrt{m}} + 2\gamma^m\sqrt{m}\right) + \frac{\gamma^c}{\lambda^c}\|\overline{G(v_S^{\mathrm{s}})}\| &&\|v_S\| \leq 3\sqrt{m} \\
&\leq \frac{1}{\lambda^m}\left(\frac{3}{\sqrt{m}} + \frac{1}{\sqrt{m}}\right) + \frac{1}{3}\|\overline{G(v_S^{\mathrm{s}})}\| &&\gamma^m \leq \frac{1}{2m},\ \gamma^c \leq \lambda^c/3 \\
&\leq 1 &&\lambda^m \geq 9/\sqrt{m}
\end{aligned}
$$

We next want to show that $\nabla F_S(\widetilde{w}) = 0$, observe that because the function $F_S$ is strictly convex, it also implies that $\widetilde{w}$ is the unique minimizer. To prove $\nabla F_S(\widetilde{w}) = 0$ we show that the following two equations hold:

$$\nabla h_S^\zeta(\widetilde{w^c}) = 0. \tag{24}$$

and,

$$\nabla p(\widetilde{w}) = \left(-\gamma^c\overline{G(v_S^{\mathrm{s}})}, \gamma^m v_S^{\mathrm{s}}\right) \qquad \text{and} \qquad p(\widetilde{w}) > 0. \tag{25}$$

Given Equations (24) and (25), we only need to plug in the differential of $p$ at $\widetilde{w}$ and the result follows from a direct calculation of the derivative in Equation (22), which yields:

$$\nabla F_S(\widetilde{w}) = \left(-\gamma^c\overline{G(v_S^{\mathrm{s}})} + \lambda^c\widetilde{w^c}, -\tfrac{1}{m}v_S + \gamma^m v_S^{\mathrm{s}} + \lambda^m\widetilde{w^m}\right) = (0,0)$$

**Proof of Equation (24)** Indeed, since $G(v_S^{\mathrm{s}}) \notin W_{z_i}$, we have by the definition of the code that for every $x \in W_{z_i}$, $\langle G(v_S^{\mathrm{s}}), \overline{x}\rangle < 1 - \frac{\rho}{2}$. Therefore, for all $j \in [m]$,

$$
\begin{aligned}
\max_{x \in W_{z_j}} \langle\widetilde{w^c}, \overline{x}\rangle &= \frac{\gamma^c}{\lambda^c} \max_{x \in W_{z_j}} \langle\overline{G(v_S^{\mathrm{s}})}, \overline{x}\rangle && \textit{Equation (23)} \\
&< \frac{\gamma^c}{\lambda^c}\left(1 - \frac{\rho}{2}\right) && \langle\overline{G(v_S^{\mathrm{s}})}, \overline{x}\rangle < 1 - \frac{\rho}{2} \\
&\leq \zeta\left(1 - \frac{\rho}{2}\right) && \textit{Equation (19) }\left(\zeta \geq \frac{\gamma^c}{\lambda^c}\right)
\end{aligned}
$$

and therefore, $h_S^\zeta(\widetilde{w^c}) = \zeta\left(1 - \frac{\rho}{2}\right)$, and by the definition of Feldman's function Equation (24) holds.

**Proof of Equation (25)** By the definition of $p$ this amounts to showing that $v_S^{\mathrm{s}}$ is the unique maximizer in the term, and that its corresponding value is nonnegative. Namely for every $v \in \{-1, 1\}^k$, distinct from $v_S^{\mathrm{s}}$:

$$\gamma^m\langle v_S^{\mathrm{s}}, \widetilde{w^m}\rangle - \gamma^c\langle\overline{G(v_S^{\mathrm{s}})}, \widetilde{w^c}\rangle > \max\left\{\gamma^m\langle v, \widetilde{w^m}\rangle - \gamma^c\langle\overline{G(v)}, \widetilde{w^c}\rangle, 0\right\}. \tag{26}$$

Let $v \neq v_S^s$, then there exists at least one coordinate $i$ such that $v(i) \neq v_S^s(i)$. Because $v_S^s(j) \cdot \widetilde{w^m}(j)$ is positive for all $j$, we obtain

$$
\begin{aligned}
\langle v_S^s, \widetilde{w^m} \rangle - \langle v, \widetilde{w^m} \rangle &\geq \left( v_S^s(i) - v(i) \right) \widetilde{w^m}(i) \\
&= 2 v_S^s(i) \widetilde{w^m}(i) && v(i) = -v_S^s(i) \\
&= \frac{2 v_S^s(i)}{\lambda^m} \left( \frac{1}{m} v_S(i) - \gamma^m v_S^s(i) \right) && \text{Equation (23)} \\
&= \frac{2}{\lambda^m} \left( \frac{1}{m} |v_S(i)| - \gamma^m \right) && v_S(i) v_S^s(i) = |v_S(i)| \\
&\geq \frac{2}{\lambda^m} \left( \frac{1}{m} - \gamma^m \right) && |v_S(i)| \geq 1 \\
&\geq \frac{1}{m \lambda^m} && \text{Equation (19)} \left( \gamma^m \leq \frac{1}{2m} \right)
\end{aligned}
\tag{27}
$$

On the other hand,

$$
\begin{aligned}
\left| \langle \overline{G(v_S^s)}, \widetilde{w^c} \rangle - \langle \overline{G(v)}, \widetilde{w^c} \rangle \right| &\leq \left\| \overline{G(v_S^s)} - \overline{G(v)} \right\| \| \widetilde{w^c} \| \\
&\leq 2 \| \widetilde{w^c} \| && \| \overline{G(v_S^s)} \| = \| \overline{G(v)} \| = 1 \\
&\leq \frac{2 \gamma^c}{\lambda^c} && \text{Equation (23)}
\end{aligned}
$$

Combining with Equation (27),

$$
\gamma^m \langle v_S^s, \widetilde{w^m} \rangle - \gamma^c \langle \overline{G(v_S^s)}, \widetilde{w^c} \rangle \geq \gamma^m \langle v, \widetilde{w^m} \rangle - \gamma^c \langle \overline{G(v)}, \widetilde{w^c} \rangle + \frac{\gamma^m}{m \lambda^m} - \frac{2(\gamma^c)^2}{\lambda^c}
\tag{28}
$$

To obtain nonnegativity of the term, a similar calculation shows that for a choice $v = 0$ and setting $G(0) = 0$:

$$
\gamma^m \langle v_S^s, \widetilde{w^m} \rangle - \gamma^c \langle \overline{G(v_S^s)}, \widetilde{w^c} \rangle \geq \frac{\gamma^m}{2m \lambda^m} - \frac{(\gamma^c)^2}{\lambda^c} = \frac{1}{2} \left( \frac{\gamma^m}{m \lambda^m} - \frac{2(\gamma^c)^2}{\lambda^c} \right).
\tag{29}
$$

Our choice of parameters yields

$$
\begin{aligned}
\frac{2(\gamma^c)^2}{\lambda^c} &\leq \frac{2}{\lambda^c} \cdot \frac{\lambda^c}{3} \cdot \frac{\gamma^m}{9\sqrt{m}} && \gamma^c \leq \min \left\{ \frac{\gamma^m}{9\sqrt{m}}, \frac{\lambda^c}{3} \right\} \\
&= \frac{2\gamma^m}{27\sqrt{m}} \\
&< \frac{\gamma^m}{m \lambda^m} && \lambda^m < \frac{27}{2\sqrt{m}},
\end{aligned}
$$

and Equation (26) holds and in turn Equation (25). $\qquad \square$

With the proof of Claim 1 concluded, this completes the proof of the case $\varepsilon = 0$.

**The general case, $\varepsilon > 0$** We proved the statement for $w_S = w_S^\star$, the unique minimizer. We now move on to the general case and assume $\varepsilon > 0$. Let $w_S$ be an $\varepsilon$-ERM solution. By strong convexity of $F_S$ we have that:

$$
\begin{aligned}
F_S(w_S) &\geq F_S(w_S^\star) + \langle \nabla F_S(w_S^\star), w_S - w_S^\star \rangle + \frac{\lambda}{2} \| w_S - w_S^\star \|^2 \\
&\geq F_S(w_S^\star) + \frac{\lambda}{2} \| w_S - w_S^\star \|^2 && \langle \nabla F_S(w_S^\star), w_S - w_S^\star \rangle \geq 0,
\end{aligned}
$$

where the last inequality follows from first order optimality of $w_S^\star$. It follows then:

$$
\begin{aligned}
F(w_S) - F(0) &= F(w_S^\star) - F(0) + F(w_S) - F(w_S^\star) \\
&\geq F(w_S^\star) - F(0) - 7\|w_S - w_S^\star\| && \text{$F$ is 7-Lipschitz} \\
&\geq F(w_S^\star) - F(0) - 7\sqrt{\frac{2}{\lambda}\left(F_S(w_S) - F_S(w_S^\star)\right)} \\
&\geq F(w_S^\star) - F(0) - 7\sqrt{\frac{2\varepsilon}{\lambda}} \\
&\geq \min\left\{\frac{\rho}{72m^{1.5}\lambda}, \frac{\rho}{12}\right\} - 7\sqrt{\frac{2\varepsilon}{\lambda}} && \text{Equation (18).}
\end{aligned}
$$

# B. Proof of the Gradient Descent Lower Bound

The proof is an immediate corollary of the following Lemma, which generalizes to other suffix-averaged iterates, defined for any $s < T$ as

$$
w_{S,s} := \frac{1}{T-s}\sum_{t=s+1}^{T} w_t. \tag{30}
$$

**Lemma B.1.** *There exists a constant $\rho \in (0, 1/2)$ such that the following holds. Fix $m > \max\{80^2, \frac{16}{\rho^2}\}$, $T \in \mathbb{N}$, $\eta \in (0,1)$, satisfying $\eta T > \sqrt{m}$, and a suffix parameter $s < T$. Then for $d = 6m$ there exist a finite instance space $\mathcal{Z}$, a distribution $D$ over $\mathcal{Z}$ and a convex loss $f : \mathcal{W}_d \times \mathcal{Z} \to \mathbb{R}$ that is 7-Lipschitz in $w \in \mathcal{W}_d$, such that for $S \sim D^m$, if we run GD for $T$ steps, with step size $\eta$, and output the suffix-average as defined in Equation (30), then with probability at least $\frac{1}{2}$ over S,*

$$
F(w_{S,s}) - \min_{w \in \mathcal{W}_d} F(w) \geq \min\left\{\frac{\rho^2}{32}\sqrt{\frac{(1 - 1/\sqrt{2})\,\eta T}{30\,m^{3/2}}}, \ \frac{\rho}{8\sqrt{3}}\right\}.
$$

**Proof.** For a fixed $m > \max\{80^2, \frac{16}{\rho^2}\}$, set $k = 2m$ and let $d = 3k \in \Theta(m)$. We will decompose the parameter vector and denote

$$
w = (w^c, w^m) \in \mathbb{R}^{2k+k},
$$

where $w^c \in \mathbb{R}^{2k}$ and $w^m \in \mathbb{R}^k$. For our construction, for each $i \in [k]$ we define the following loss function, parameterized by $\zeta, \gamma^c, \lambda^m, \lambda^c \leq 1$ and $\gamma^m \leq \frac{1}{\sqrt{k}}$

$$
f(w, i) := h^\zeta(w^c, i) - \langle w^m, \delta_i\rangle + \max\{p(w), 0\} + \frac{\lambda^m}{2}\|w^m\|^2 + \frac{\lambda^c}{2}\|w^c\|^2, \tag{31}
$$

where, $h^\zeta(w, i)$ is Feldman's function in $\mathbb{R}^{2k}$, with an asymptotically good code $G = G_k$ as defined in Equation (14):

$$
\delta_i(j) = \begin{cases} 1/m - 2 & i = j \\ 1/m & i \neq j \end{cases}, \qquad p(w) := \max_{v \in \{-1,1\}^k}\left\{\gamma^m\langle v, w^m\rangle - \gamma^c\langle\overline{G(v)}, w^c\rangle\right\}.
$$

One can observe that each term is convex, moreover $\|\delta_i\| \leq 2$ and $\gamma^m\sqrt{k} \leq 1$, so overall $f$ is 7-Lipschitz in the unit ball. Then, we let $D$ be a uniform distribution over $[k]$. In our setting, we sample $m$ indices $z_1, z_2, \ldots, z_m$ independently and uniformly from the set $\{1, 2, \ldots, 2m\}$, and denote by $m_i$ the number of times the index $i$ is chosen. We set for a sample $S$ the vectors:

$$
v_S(i) = \begin{cases} 1 - 2m_i & i \in S \\ 1 & i \notin S \end{cases} \qquad ; \qquad v_S^s(i) = \begin{cases} -1 & i \in S \\ 1 & i \notin S \end{cases}
$$

Applying Lemma A.2 to the random variable $X_i = \delta_i$ yields that, with probability at least $1/2$,

$$
\|v_S\| = \|v_S - m \cdot 0\| = \left\|v_S - \frac{m}{k}\sum_{i=1}^{k}\delta_i\right\| = \left\|\sum_{i=1}^{m}\delta_{z_i} - m \cdot \mathop{\mathbb{E}}_{i \sim D}[\delta_i]\right\| \leq 3\sqrt{m} \tag{32}
$$

We throughout assume this event happened, and analyze the generalization error under this event. We now make the following claim

**Claim 2.** *Suppose we choose the parameters of $f(w, i)$ in the following regime:*

$$\frac{18}{\sqrt{2m}} \leq \lambda^m < \frac{18}{\sqrt{m}}, \qquad \gamma^m < \frac{1}{m} - \frac{\lambda^m}{18\sqrt{m}}, \quad \gamma^c \leq \min\left\{\sqrt{\frac{\gamma^m}{30\sqrt{m}\eta T}}, \frac{\lambda^c}{\sqrt{3}}\right\}, \quad \zeta \geq \frac{\gamma^c}{\lambda^c}, \tag{33}$$

*and suppose that $\eta T > \sqrt{m}$. Then for the update rule of GD over the empirical risk with sample S is given by:*

$$w_t^c = \frac{\gamma^c}{\lambda^c}\left(1 - (1 - \eta\lambda^c)^{t-1}\right)\overline{G(v_S^s)}, \quad w_t^m = (1 - \eta\lambda^m)^{t-1}\frac{\eta}{m}v_S + \frac{1 - (1 - \eta\lambda^m)^{t-1}}{\lambda^m}\left(\frac{1}{m}v_S - \gamma^m v_S^s\right) \tag{34}$$

We defer the proof of the claim to Section B.1 and let us first observe how it entails the statement. First, observe that for every $w$:

$$F(w) - F(0) \geq h_D^\zeta(w^c) - \zeta\left(1 - \frac{\rho}{2}\right),$$

since $\mathbb{E}[\delta_i] = 0$. Therefore, it is enough to show that for the output $w_{S,s} = (w_{S,s}^c, w_{S,s}^m)$ we have that:

$$h_D^\zeta(w_{S,s}^c) - \zeta\left(1 - \frac{\rho}{2}\right) \geq \min\left\{\frac{\rho^2}{32}\sqrt{\frac{(1 - 1/\sqrt{2})\eta T}{30\,m^{3/2}}}, \frac{\rho}{8\sqrt{3}}\right\}. \tag{35}$$

Indeed, we have that:

$$\begin{aligned}
\langle w_{S,s}^c, \overline{G(v_S^s)}\rangle &= \frac{1}{T - s}\sum_{t=s+1}^{T}\langle w_t^c, \overline{G(v_S^s)}\rangle \\
&= \frac{1}{T - s}\sum_{t=s+1}^{T}\frac{\gamma^c}{\lambda^c}\left(1 - (1 - \eta\lambda^c)^{t-1}\right)\|\overline{G(v_S^s)}\|^2 && \textit{Equation (34)} \\
&\geq \frac{\gamma^c}{T\lambda^c}\sum_{t=1}^{T}\left(1 - (1 - \eta\lambda^c)^{t-1}\right) \\
&\geq \frac{\gamma^c}{\lambda^c}\left(1 - \frac{1 - (1 - \eta\lambda^c)^T}{\lambda^c\eta T}\right) \\
&\geq \frac{\gamma^c}{\lambda^c}\left(1 - \frac{1}{\lambda^c\eta T}\right) && 0 < \eta\lambda^c < 1
\end{aligned}$$

Now we use the definition of Feldman's function, and the fact that for $i \sim D$ we have $G(v_S^s) \in W_i$ with probability at least $1 - \frac{m}{k} \geq \frac{1}{2}$ and we obtain:

$$h_D^\zeta(w_{S,s}^c) - \zeta(1 - \frac{\rho}{2}) \geq \frac{1}{2}\left(\frac{\gamma^c}{\lambda^c}\left(1 - \frac{1}{\lambda^c\eta T}\right) - \zeta + \frac{\zeta\rho}{2}\right)$$

Setting $\zeta = \frac{\gamma^c}{\lambda^c}$ and $\lambda^c = \frac{4}{\rho\eta T}$ (note that $\lambda^c \leq 1$ since $\eta T > \sqrt{m}$ and $m > \frac{16}{\rho^2}$), we obtain:

$$\begin{aligned}
h_D^\zeta(w_{S,s}^c) - \zeta(1 - \frac{\rho}{2}) &\geq \frac{1}{2}\left(\frac{\gamma^c\rho}{2\lambda^c} - \frac{\gamma^c}{(\lambda^c)^2\eta T}\right) && \zeta = \frac{\gamma^c}{\lambda^c} \\
&= \frac{\gamma^c\rho^2\eta T}{32},
\end{aligned}$$

and Equation (35) follows by choosing

$$\lambda^m = \frac{18}{\sqrt{2m}}, \qquad \gamma^m = \frac{1}{m} - \frac{1}{\sqrt{2m}}, \tag{36}$$

$$\gamma^c = \min\left\{\sqrt{\frac{\gamma^m}{30\sqrt{m}\eta T}}, \frac{4}{\rho\eta T\sqrt{3}}\right\} = \Theta\left(\min\left\{\sqrt{\frac{1}{m^{1.5}\eta T}}, \frac{1}{\eta T}\right\}\right).$$

$\blacksquare$

## B.1. Proof of Claim 2

We are thus left to show that Equation (34) holds under the regime of Equation (33), which will complete the proof. We prove it by induction. The case $t = 1$ can be verified by observing that the empirical risk for a sample $S$ is given by:

$$F_S(w) = h_S^\zeta(w^c) - \frac{1}{m}\langle w^m, v_S\rangle + \max\{p(w), 0\} + \frac{\lambda^m}{2}\|w^m\|^2 + \frac{\lambda^c}{2}\|w^c\|^2.$$

Then we note that $\nabla h_S^\zeta(0) = 0$, and $p(w_0) = p(0) = 0$ which entails that $0 \in \partial \max\{p(w_0), 0\}$. One can show that our bounds on the learning rate and $m$ ensures that no projection is involved, then $w_1^c = 0$ and $w_1^m = \frac{\eta}{m}v_S$, as required. We next assume Equation (34) holds for $t$ and we prove it for $t + 1$. To prove the claim we need to show that:

$$\nabla h_S^\zeta(w_t^c) = 0 \tag{37}$$

$$\nabla p(w_t) = \left(-\gamma^c\overline{G(v_S^s)}, \gamma^m v_S^s\right), \qquad \text{and} \qquad p(w_t) > 0 \tag{38}$$

Once we obtain these two facts, the result follows from the induction hypothesis and the following computation:

$$
\begin{aligned}
w_{t+1}^m &= w_t^m - \eta\partial_{w^m}F_S(w_t)\\
&= w_t^m + \frac{\eta}{m}v_S - \eta\gamma^m v_S^s - \eta\lambda^m w_t^m\\
&= (1 - \eta\lambda^m)\, w_t^m + \frac{\eta}{m}v_S - \eta\gamma^m v_S^s && \textit{Equation (38)}\\
&= (1 - \eta\lambda^m)\left((1 - \eta\lambda^m)^{t-1}\frac{\eta}{m}v_S + \frac{1 - (1 - \eta\lambda^m)^{t-1}}{\lambda^m}(\frac{1}{m}v_S - \gamma^m v_S^s)\right) + \frac{\eta}{m}v_S - \eta\gamma^m v_S^s\\
&= (1 - \eta\lambda^m)^t\frac{\eta}{m}v_S + \frac{(1 - \eta\lambda^m) - (1 - \eta\lambda^m)^t}{\lambda^m}(\frac{1}{m}v_S - \gamma^m v_S^s) + \frac{\eta}{m}v_S - \eta\gamma^m v_S^s\\
&= (1 - \eta\lambda^m)^t\frac{\eta}{m}v_S + \frac{1 - (1 - \eta\lambda^m)^t}{\lambda^m}\left(\frac{1}{m}v_S - \gamma^m v_S^s\right)
\end{aligned}
$$

and

$$
\begin{aligned}
w_{t+1}^c &= w_t^c - \eta\partial_{w^c}F_S(w_t)\\
&= w_t^c + \gamma^c\eta\overline{G(v_S^s)} - \lambda^c\eta w_t^c && \textit{Equations (37) and (38)}\\
&= (1 - \eta\lambda^c)w_t^c + \gamma^c\eta\overline{G(v_S^s)}\\
&= \frac{\gamma^c(1 - \eta\lambda^c)}{\lambda^c}\left(1 - (1 - \eta\lambda^c)^{t-1}\right)\overline{G(v_S^s)} + \gamma^c\eta\overline{G(v_S^s)}\\
&= \frac{\gamma^c}{\lambda^c}\left(1 - (1 - \eta\lambda^c)^t\right)\overline{G(v_S^s)}
\end{aligned}
$$

The norm of both terms is bounded: since by Equation (33) $\frac{\gamma^c}{\lambda^c} \leq 1/\sqrt{3}$ and therefore $\|w_{t+1}^c\| \leq 1/\sqrt{3}$ and

$$
\begin{aligned}
\|w_{t+1}^m\| &\leq \frac{\eta}{m}\|v_S\| + \frac{1}{\lambda^m}\left(\frac{1}{m}\|v_S\| + \gamma^m\|v_S^s\|\right)\\
&\leq \frac{3\eta}{\sqrt{m}} + \frac{1}{\lambda^m}\left(\frac{3}{\sqrt{m}} + \frac{\sqrt{2}}{\sqrt{m}}\right) && \textit{Equation (32)}\\
&\leq \frac{3}{\sqrt{m}} + \frac{\sqrt{2m}}{18}\cdot\frac{3 + \sqrt{2}}{\sqrt{m}} && \textit{Equation (33)}\\
&\leq \sqrt{\frac{2}{3}} && m > 80^2
\end{aligned}
$$

In turn, $\|w_{t+1}\|^2 = \|w_{t+1}^c\|^2 + \|w_{t+1}^m\|^2 \leq \frac{1}{3} + \frac{2}{3} = 1$, and the projection onto the unit ball is inactive, and the proof is complete. We are thus left with showing Equations (37) and (38).

**Proof of Equation (37)**  We begin by showing that $\nabla h_S^\zeta(w_t^c) = 0$. Indeed, notice that for every $x \in W_{z_i}$, since $G(v_S^s) \notin W_{z_i}$, we have by the definition of the code that $\langle G(v_S^s), \overline{x} \rangle < 1 - \frac{\rho}{2}$. In particular, for $w_t^c$, by the induction hypothesis and the assumption $\zeta \geq \frac{\gamma^c}{\lambda^c}$:

$$
\begin{aligned}
\max_{x \in W_{z_i}} \langle w_t^c, \overline{x} \rangle &= \max_{x \in W_{z_i}} \left\{ \frac{\gamma^c}{\lambda^c}(1 - (1 - \eta\lambda^c)^{t-1})\langle \overline{G(v_S^s)}, \overline{x} \rangle \right\} && \text{Equation (34)} \\
&< \frac{\gamma^c}{\lambda^c}\left(1 - \frac{\rho}{2}\right) && G(v_S^s) \notin W_{z_i} \\
&\leq \zeta\left(1 - \frac{\rho}{2}\right) && \zeta \geq \frac{\gamma^c}{\lambda^c}
\end{aligned}
$$

Therefore, $\nabla h^\zeta(w_t^c, z_i) = 0$.

**Proof of Equation (38)**  Next, we want to show that

$$
v_S^s = \arg \max_{v \in \{-1,1\}^k} \left\{ \gamma^m \langle v, w_t^m \rangle - \gamma^c \langle \overline{G(v)}, w_t^c \rangle \right\}, \qquad \text{and} \qquad p(w_t) > 0 \tag{39}
$$

and that $v_S^s$ is the unique maximizer. This will establish Equation (38). To see Equation (39), first note that $v_S^s$ indeed maximizes the first term. Now, replace $v_S^s$ by any $v \in \{-1,1\}^k$ that differs from $v_S^s$ in at least one coordinate (say, $i$). This will decrease the first term by at least $2\gamma^m |w_t(i)|$:

$$
\begin{aligned}
2\gamma^m |w_t^m(i)| = 2\gamma^m \cdot &\left( (1 - \eta\lambda^m)^{t-1} \frac{\eta}{m}|v_S(i)| + \frac{1 - (1 - \eta\lambda^m)^{t-1}}{\lambda^m}\left(\frac{1}{m}|v_S(i)| - \gamma^m\right) \right) \\
&\geq 2\gamma^m \cdot \left( \frac{1 - (1 - \eta\lambda^m)^{t-1}}{\lambda^m}\left(\frac{1}{m}|v_S(i)| - \gamma^m\right) \right) \\
&\geq \frac{\gamma^m}{9\sqrt{m}}\left(1 - (1 - \eta\lambda^m)^{t-1}\right) && \text{Equation (33), } 1 \leq |v_S(i)| \\
&\geq \frac{\gamma^m}{9\sqrt{m}}\left(1 - e^{-(t-1)\eta\lambda^m}\right) && 1 - x < e^{-x} \\
&\geq \frac{\gamma^m}{9\sqrt{m}} \min\left\{ \frac{(t-1)\eta\lambda^m}{2}, 1 - e^{-1} \right\} \\
&\geq \min\left\{ \frac{\gamma^m(t-1)\eta}{m\sqrt{2}}, \frac{\gamma^m(1 - e^{-1})}{9\sqrt{m}} \right\}.
\end{aligned}
$$

The last inequality uses $\lambda^m > \frac{18}{\sqrt{2m}}$, and in the one before last inequality we used that $1 - e^{-x} \geq \min\{x/2, 1 - e^{-1}\}$ for all $x \geq 0$. The second term, by Cauchy-Schwartz, can increase by at most

$$
\begin{aligned}
2\gamma^c|\langle \overline{G(v)}, w_t^c \rangle| \leq 2\gamma^c \|w_t^c\| &= 2\frac{(\gamma^c)^2}{\lambda^c}\left(1 - (1 - \eta\lambda^c)^{t-1}\right) \\
&= 2\eta(\gamma^c)^2 \frac{1 - (1 - \eta\lambda^c)^{t-1}}{1 - (1 - \eta\lambda^c)} \\
&= 2\eta(\gamma^c)^2 \sum_{t'=0}^{t-2} (1 - \eta\lambda^c)^{t'} \\
&< 2(\gamma^c)^2\eta(t-1)
\end{aligned}
$$

Finally, the conditions $\gamma^c \leq \sqrt{\frac{\gamma^m}{30\sqrt{m}\eta T}}$ and $\eta T > \sqrt{m}$ ensures that Equation (39) indeed holds (one can show that a similar calculation also holds by taking $v = 0$, hence $p(w_t) > 0$).

