# OpenReview forum: "All ERMs Can Fail in Stochastic Convex Optimization Lower Bounds in Linear Dimension"
_ICML.cc/2026/Conference — ICML 2026 regular_

### Official Review · Reviewer_cXcy · 2026-03-03

**Soundness:** 4
**Presentation:** 4
**Significance:** 3
**Originality:** 3
**Overall Recommendation:** 5
**Confidence:** 4

**Summary:**

This paper provides an instance of stochastic convex optimization problem where all approximate ERM with error below particular order have at least a generalization gap of constant order, which also has an implication on the generalization bound of GD trained with sufficiently long horizon, thereby addressing some of the open problems appeared in prior literature.

**Compliance With Llm Reviewing Policy:**

Affirmed.

**Final Justification:**

The authors have addressed my questions in the rebuttal. I keep the positive rating.

**Key Questions For Authors:**

1. In the paragraph from line 71 to line 81 (left), I could not see how the discussion eventually arrives at a conclusion "Our study show that their success is independent from some generic property of the empirical minimizer, and succeed to learn even when the Guess & Check algorithm fails in SCO". The issue is with the Guess & Check algorithm, I did see any concrete results regarding G&C discussed either in the text before or in the following sections. So to me, this "Guess & Check algorithm fails in SCO" came out of nowhere. Can author clarify this discussion? Also, the mentioning of G&C reads as if the following sections are going to discuss G&C but they do not?

2. Does the results provide any insights as to why ERM does not fail in practice? Should be it be that the typical problems in practice does not possess a similar structure like the problem constructed in the paper?

3. line 249 to line 250, missing a reference in "Recall that an $O(1/\sqrt{m})$-optimal solution can always be found using SGD."?

**Limitations:**

yes

**Strengths And Weaknesses:**

Strength:
1. The paper is very well-written with good literature reviews. The bounds in theorems are clearly explained relative to existing results. The construction of the strongly-convex function might be of interest to whom working on problems of similar kinds.
2. Understanding the limitation of ERM, and some optimization algorithms like GD are important research topics in ML. The paper makes clear contributions to those questions by addressing open problems, explaining the proof techniques, and providing discussions on future directions.

Weakness:
See questions

---

> ### Author Rebuttal · Authors · 2026-03-30
>
> We thank the reviewer for the careful reading, insightful questions, and positive evaluation, we truly appreciate it.
>
> ---
>
> > **“The issue is with the Guess & Check algorithm, I did [not] see any concrete results regarding G&C discussed”**
>
> You are right, we should have discussed this further after the main result and explain, specifically, how it relates to G&C.  Let us clarify.
> Guess & Check refers to an algorithm that picks randomly a solution that minimizes the risk. In particular it is an ERM.
> In our work, we construct a setting where there is a unique empirical minimizer hence all ERMs behave the same. In other words, the G&C algorithm picks deterministically the single unique ERM. By our result, it will have a bad generalization error because all ERM(s) behave badly.
>
> We hope this clarifies the connection to G&C. We will revisit G&C after the main result and discuss this implication.
>
> ---
>
> > **“Does the results provide any insights as to why ERM does not fail in practice?”**
>
> This is an excellent question. Our results are worst-case, so they should not be interpreted as saying that ERM typically fails in practice. Rather, the takeaway is that the success of learning in convex problems cannot, in general, be explained by a generic property shared by all empirical minimizers.
>
> We believe the results suggest two possible interpretations for practice. First, it could be that in practical problems “typical” fitting solutions may behave much better.
>
> Second, and perhaps more interesting, practical optimization methods are not arbitrary ERMs: they come with implicit or explicit bias, stability, initialization effects, regularization, early stopping, etc. It is likely that such algorithmic bias *does matter*, because worst-case success cannot be attributed to the empirical minimizer set itself.
>
> ---
>
> > **“line 249 to line 250, missing a reference”**
>
> Thank you, this is a great catch. We will add an appropriate reference for the standard SGD guarantee in stochastic convex optimization.
>
> ---
>
> Once again, we sincerely thank the reviewer for the constructive feedback, we believe these clarifications will strengthen the paper.

---

> > ### Author Rebuttal · Reviewer_cXcy · 2026-03-31
> >
> > I keep my score

---

### Official Review · Reviewer_PpBG · 2026-03-11

**Soundness:** 4
**Presentation:** 4
**Significance:** 4
**Originality:** 3
**Overall Recommendation:** 5
**Confidence:** 5

**Summary:**

This paper demonstrates an instance of a strongly convex and Lipschitz loss, in dimension linear in the number of samples, such that the only ERM has poor generalization error (generalization error Omega(1) in a particular parameter regime). A similar construction of a strongly convex and Lipschitz loss, in dimension linear in the number of samples, s is provided against gradient descent, again with generalization error Omega(1) in a particular parameter regime. The proof idea builds on the construction of Feldman (2016).

**Compliance With Llm Reviewing Policy:**

Affirmed.

**Final Justification:**

It is an interesting and strong paper

**Key Questions For Authors:**

My main question is: is it true that the same loss can provide the lower bound against both ERM and GD, specifically Lemma A.1 and B.1? It seems like the loss takes the same form and the parameter regimes for both results are consistent. This would even further strengthen the results, establishing both algorithms fail on the same convex loss. Am I overlooking something?

Minor points:

-Why is $m \le k/2$ used here: "Notice that, given a random, uniform finite sample $z_1, \ldots, z_m \in [k]^m$, when $m \le k/2$, ..."? I think this just requires $m \le k-1$ (take $v$ with -1 in the $\le m$ coordinates corresponding to the $z$'s and +1 in the $\ge k-m$ other coordinates).

-Also, in 585-586 I think the definition should read $\delta_i(j)=1/m-1$ for $i=j$ so that $E[\delta_z]=0$, if I'm not mistaken?

-There is conditioning on the event $||v_S|| \le 3\sqrt{m}$ which holds with probability >=1/2 and that $G(v_S^s) \in W_z$ with probability >=1/2. These probabilities are not obviously independent, I think the constants should be adjusted to get two probabilities that sum to <1? (and anyhow as its written, the probability of failure is not 1/2)

**Limitations:**

yes

**Strengths And Weaknesses:**

I believe the work is of high quality and I am strongly supportive of its publication in ICML.

The results are strong. The construction of the lower bound is not degenerate, in that it satisfies all the properties one might expect lead to learning: the loss is strongly-convex, Lipschitz loss, in dimension linear in the number of samples, yet all ERMs (specifically the only one) fails. It is worth noting dimension linear in the number of samples is a regime of major interest in high-dimensional statistics. A lower bound is provided for approximate ERM and for GD and it is compelling that there are lower bounds provided against several different methods (in fact I believe that following the proof, the same loss can provide the lower bound against both ERM and GD). The work is grounded in the literature and answers an open question from Feldman (2016), and also adds food for thought for practitioners. The proof is simple, but in a good way: the arguments are short and elegant and build nicely off the ideas of Feldman.

The presentation and writing is also excellent and I found the paper (including the appendix) a pleasure to read.

Regarding weaknesses, there are no major ones, but I found the proof sketch in 4.2, 4.3 rather hard to follow. Indeed it was easier to just read the proof in the appendix. I feel that the proof sketch in 4.2 is not much easier to understand than the full proof itself, I would recommend just writing Lemma A.1 and the major steps in its proof where 4.2 currently is, leaving details for the appendix, and explain briefly how the GD proof follows. It would also be nice to elaborate more on the proof of Feldman in 4.1, as these ideas are very important to follow this paper's proofs. I also think the proof is somewhat straightforward, and mostly builds off of the ideas from Feldman (2016), that said this does not detract from the strength of the lower bounds. Last, a few small points are below and in my questions, but they also do not detract from the results and quality of the work.

A few small items.

-Several citations are missing on implicit bias of gradient descent (e.g. Gunasekar, Suriya, et al, "Implicit bias of gradient descent on linear convolutional networks", NeurIPS 2018), though these citations are not directly related to this work.

-there should be a comma on eq (19), lines 619-620, and the next sentence should not be capitalized.

---

> ### Author Rebuttal · Authors · 2026-03-30
>
> We thank the reviewer for the very careful reading, the strong support, and the many helpful suggestions.
>
> ---
>
> > **“is it true that the same loss can provide the lower bound against both ERM and GD, specifically Lemma A.1 and B.1?”**
>
> Thank you for this very insightful question.For the long-horizon GD lower bound, this is already implicit in the paper, since the corollary follows by combining the ERM construction with the fact that GD becomes an $\epsilon$-ERM once the optimization error is sufficiently small.
>
> For the more refined GD lower bound, we use ”almost” the same function but with slightly different hyperparameter choice. We don’t know if this is an artifact of our proof or essential. In fact, even just looking at GD our construction depends on the choice of  hyperparameters. So an interesting, related, question is if the same loss can provide the lower bound against GD independent of hyperparameter choice.
>
> ---
>
> > **“I found the proof sketch in 4.2, 4.3 rather hard to follow. Indeed it was easier to just read the proof in the appendix.”**
>
> Thank you very much for this constructive feedback. We wanted to incorporate an intuitive proof sketch in a simplified setting (not insisting on linear dimension, and not focusing on approximate-ERM). That being said, the current exposition may contain some unnecessary intuitions and details.
>
> We will follow the reviewer’s suggestion. We will shorten the preliminary discussion in section 4.2 and instead write a simplified version of Lemma A.1 that includes only the special case $\epsilon=0$, and allows $d=m^2$ (hence doesn’t require Lemma A.2), along with its formal proof as space permits.
>
> ---
>
> > **“Several citations are missing on implicit bias of gradient descent (e.g. Gunasekar, Suriya, et al, "Implicit bias of gradient descent on linear convolutional networks", NeurIPS 2018), though these citations are not directly related to this work.”**
>
> Thank you for pointing this out. We will add the suggested citations and further works.
>
> ---
>
> > **“there should be a comma on eq (19), lines 619-620, and the next sentence should not be capitalized.”**
>
> Thank you, we will fix the punctuation issue around Eq. (19) and the capitalization in the following sentence.
>
> ---
>
> > **“Why is $m\leq k/2$ used here”**
>
> Thank you for this observation. The reason we use $m \le k/2$ there is not only to guarantee existence of a codeword outside the sampled sets, but we also need to ensure a constant population gap. This requires that samples who did not appear during training are likely to appear during testing and thus $m=k-1$ is not sufficient. Notice that this is already the case in Feldman’s construction.
>
> ---
>
> > **“Also, in 585-586 I think the definition should read”**
>
> Thank you again for your observation. We would like to clarify that the definition is in fact correct.
>
> In particular, $\delta_i$ must take the value $-2$ (and not $-1$) in order for $\mathbb{E}[\delta_i] = 0$. This follows from the fact that $k = 2m$ (rather than $m$), and the expectation is taken over $z_i \in ${$z_1, z_2, \dots, z_k$}. Under this distribution, setting $\delta_i(j) = 1/m - 2$ for $i = j$ ensures that the mean is exactly zero.
>
> We will include a short explicit calculation in the revision to make this fully clear and avoid any confusion.
>
> ---
>
> > **“There is conditioning on the event”**
>
> Thank you again for your observation. We would like to clarify that this interpretation is not correct.
>
> We condition *only* on the event $\|v_S\| \le 3\sqrt{m}$. The fact that $G(v^s_S) \in W_z$ holds with probability at least $1/2$ is not used as a conditioning event. Instead, this probability is used only to lower bound $h_D^\zeta$, i.e., it is used inside the expectation calculation, and not as part of the conditioning.
>
> We will revise this part to make this distinction more explicit.
>
> ---
>
> Once again, we sincerely thank the reviewer for the strong support and the detailed suggestions. We believe these clarifications and fixes will make the paper clearer and stronger.

---

> > ### Author Rebuttal · Reviewer_PpBG · 2026-04-01
> >
> > Thank you for the authors for the reply and explaining several of my technical confusions! My questions are well-answered and I have no further questions. I continue to believe this paper is of strong quality and I am strongly supportive of its publication in ICML.

---

### Official Review · Reviewer_kHZZ · 2026-03-13

**Soundness:** 3
**Presentation:** 2
**Significance:** 3
**Originality:** 3
**Overall Recommendation:** 4
**Confidence:** 1

**Summary:**

This paper studies stochastic optimization. It is shown that there are instances where learning is possible, yet ERM/approximate ERM are likely to overfit. In particular, a $\Omega(\eta T/(m\sqrt{m})$ generalization lower bound is derived for gradient descent with step-size $\eta$, where $m$ is the sample size. A consideration in this work is that dimension is linear in the number of samples (contrasting prior works which derive such generalization gaps in dimensions exponential in the sample size), answering the open problem of Feldman (2016).

**Compliance With Llm Reviewing Policy:**

Affirmed.

**Key Questions For Authors:**

n/a

**Limitations:**

yes

**Strengths And Weaknesses:**

- The result seems correct and non-trivial, and the lower bound construction is interesting. The main result answers an open problem.

- The significance of the achievement is hard for me to gauge since I rarely think about statistical guarantees regarding generalization; it's really not my sub-field. The result seems reasonably non-trivial to prove, at least.

- The writing feels unpolished in general and is often unconventional (e.g. line 60, "... an active, highly intense field of study")
  - The precise meaning of "failing to generalize" is never actually given but the entire paper revolves around this. I think it can be inferred from the sample complexity definition in Section 2 but this should be spelled out more explicitly in the main text if it's going to play such a central role.

---

> ### Author Rebuttal · Authors · 2026-03-30
>
> We thank the reviewer for the feedback and for taking the time to read our paper.
>
> ---
>
> > **“The writing feels unpolished in general and is often unconventional (e.g. line 60, "... an active, highly intense field of study")”**
>
> Thank you for this comment. We will revise the writing to improve clarity and ensure a more polished and conventional style throughout the paper. We will change line 60 to “an active area of research”.
>
> ---
>
> > **“The precise meaning of "failing to generalize" is never actually given but the entire paper revolves around this”**
>
> Thanks for pointing this out. First, we wish to clarify that a formal definition of the learner’s objective appears in Eq. 3, and the expression “failing to generalize” does not appear in any formal result.
>
> We remark on some occasions that certain algorithms “fail to generalize” and this refers to the situation where the output parameter that minimizes the empirical risk performs poorly on the population loss (i.e. eq 3 is not met). In particular, the learned parameter achieves near-constant (non-informative) loss on the population distribution despite fitting the training data well. We are happy to clarify any point further and would greatly appreciate any additional questions.

---

> > ### Author Rebuttal · Reviewer_kHZZ · 2026-03-31
> >
> > Great, thanks. After reading the other rebuttals and replies, I'm more confident that the paper has strong results, although I do get the impression that clarity of presentation might be a persistent issue. Taken together I will keep my current (positive) score.

---

### Official Review · Reviewer_FUwg · 2026-03-24

**Soundness:** 2
**Presentation:** 2
**Significance:** 4
**Originality:** 3
**Overall Recommendation:** 4
**Confidence:** 4

**Summary:**

The paper studies generalizability of empirical risk minimization (ERM) or equivalently, sample average approximation (SAA), in stochastic convex optimization. Theorem 3.1 identifies scenarios where ERM leads to at least a constant generalization error. Theorem 3.2 further discusses a different lower bound when the modulus of strong convexity is subject to some larger lower bounds. Corollary 3.3 and Theorem 3.4 then extends the analysis to gradient descent algorithms. The paper claims to have answered an open question posed by Feldman (2016): whether a construction of ERM exists in dimension that is linear in the sample size such that the solutions are (unique and) overfitting.

**Compliance With Llm Reviewing Policy:**

Affirmed.

**Final Justification:**

The analysis and proof are overall correct, despite several issues with regard to rigor and presentation, as is pointed out in the previous comments of mine. Nonetheless, I believe that the reason for accepting this paper outweighs the weaknesses. I would encourage the authors to go through the paper to fix the raised issues so that others can build on their work to obtain correct and useful results.

**Key Questions For Authors:**

Q1. Related to both W1, can the authors thoroughly discuss their results relative to Example 5.21 of Shapiro et al. (2021) and Guigues et al. (2017)?

Q2. Related to W2, can the authors carefully and explicitly state the conditions/assumptions for their results?

Q3. Can the authors fix the issues about proof rigor/correctness mentioned in W3~8?

**Limitations:**

Yes

**Strengths And Weaknesses:**

Strengths:

S1. The paper targets an open question in literature. The significance appears excellent to me so long as the analysis is actually correct (See Weaknesses below).

S2. ERM and GD are widely useful schemes in machine learning; theoretical advancement/understanding in this area can be valuable.

Weaknesses:

W1. The paper can benefit much from a more thorough discussion of how the results can delineate from existing findings in the literature. For instance, in Euclidean space with $d$ growing linearly with $m$, constant lower bounds for ERM (or equivalently sample average approximation) in convex stochastic optimization seems immediately implied by Example 5.21 of Shapiro et al. (2021) and Guigues et al. (2017). These previous results, though focus on non-regularized SAA, seem directly convertible to Tikhonov-regularized SAA formulations particularly in the strongly convex case. Meanwhile, solutions to exact SAA can also be viewed as approximate solutions to Tikhonov-regularized SAA, which are also relevant to the results of this current manuscript.

W2. The preliminaries and assumptions (e.g., convexity and Lipschitz conditions) are discussed for treating $f$ as a deterministic function (with no uncertain parameters). When these conditions are later imposed on $f(\cdot,z)$, it is misleading and hard to evaluate their appropriateness/generality.

W3. The analysis is not sufficiently rigorous. E.g., the discussions do not assume differentiability and discuss subdifferential in multiple locations, indicating the focus of nonsmooth convex optimization. However, in the meantime, gradients are used directly, such as in Line 746, Line 672—674. Even if differentiability is additionally imposed, the use of gradients of F_S as defined in (22) is also problematic.

W4. Related to the previous point, what “GD” actually means in terms of its oracle should be explicitly discussed. Do you assume a subgradient selection or an unbiased estimator of a subgradient of $f$?

W5. Line 881: “One can show that our bounds on the learning rate and m ensures that no projection is involved.” Not hitting the boundary and thus not involving the projection are usually hard to ensure in stochastic optimization. This claim, while vital to the proof, is actually nontrivial to see.

W6. Proof of (37) analyzes the impacts of changing one coordinate to the decision variable. The generalization of one-coordinate to all coordinates seems very non-trivial but is skipped without a proof.

W7. Line 588, “One can observe that f is the sum of 6 convex and 1-Lipschitz functions”; it is actually hard to understand what this means and how to observe the claimed properties, as f is defined in different ways throughout this manuscript. In particular, the nearest definition involves two arguments. The function is unlikely convex in the second argument.

W8. Proof hinges on feasibility of selecting parameters/problem quantities such that  $\gamma_c \le \sqrt{\gamma_m/(200\sqrt m\,\eta T)}$, $\eta T>\sqrt m$; whether they are feasible has to be explicitly shown within the assumed ranges of quantities and parameters as in the theorem statements.

W9. The reading and evaluation of the proofs are obscured by the organization. Below are a few examples.


    9.1. The proof of Equation (24) to Equation (25) to Equation (26) is not structurally straightforward. The text moves from “flat Feldman term” to “active maximizer” to “positivity” without first stating explicitly that these are the three ingredients needed to identify the relevant (sub)gradient of $F_S$. The reader has to infer what each displayed equation is for.

    9.2. the role of Equations (25), (26), (27), (28), and (29) is not staged well. The proof zigzags through intermediate steps. In particular, equation (25) needs both identification of the maximizing sign vector, and positivity of the maximized value. But the proof first handles Equation (26) and Equation (29), and then only afterward says “Equation (26) holds and in turn Equation (25).”

    9.3. The writeup hides key proof steps inside phrases like “one can observe,” “similar calculation,” and “finally.”

Minor concerns:

1. Eq. (8), $\eta$ appears undefined.
2. The paper should clarify early on (perhaps starting in abstract) what “in linear dimension” means to avoid ambiguity.


V. Guigues, A. Juditsky, and A. Nemirovski. Non-asymptotic confidence bounds for the optimal value of a stochastic program. Optimization Methods and Software, 32(5):1033–1058, 2017.

A. Shapiro, D. Dentcheva, and A. Ruszczynski. Lectures on stochastic programming: modeling and theory. SIAM, 2021.

---

> ### Author Rebuttal · Authors · 2026-03-30
>
> We thank the reviewer for the careful reading, below we provide detailed answers to the concerns.
>
> ---
>
> > **W1+Q1**
>
> Thank you for these references.
> Regarding Example 5.21 in Shapiro et al. (2021), the corresponding loss functions are not $O(1)$-Lipschitz, (they incorporate a Gaussian noise vector whose norm scales with the dimension). We will add to the related work a discussion on such settings that are at the periphery of our setup (non-Lipschitz constructions and non-convex constructions).
>
> As for Guigues et al. (2017), thank you for this reference. Proposition 2 states that there exists a bad ERM solution. It is analogous in that respect to Feldman’s result (in fact the constructions are very similar), which we build on and discuss at length. We will discuss this work in the related work\introduction section and state that similar constructions to Feldman’s also appeared there.
>
> ---
>
> > **W2+Q2**
>
> Our preliminaries and assumptions *do* incorporate uncertain parameters (line 145: “we consider a function $f:W_d\times Z \to R$ that is convex and Lipschitz in its first argument”). In the next sentence, we recall the definition of convexity and Lipschitz. We indeed used the letter f for the recollection which may be confusing. We will revise this section and use another letter (e.g. h) when defining convexity and Lipschitness (i.e. “Recall that a function h(w) is convex and Lipschitz if…“).
>
> ---
>
> > **W3**
>
> Our result is a non-smooth construction, we do not impose differentiability, which would only strengthen the result, as our result is a lower bound.
> As stated in lines 152–155, at points where the subdifferential is unique, the function is differentiable, and then, we denote the unique element (which coincides with the derivative) with the symbol $\nabla$. This applies to the instances mentioned (Lines 672–674 and 746). We did not discuss at length the fact that a unique subdifferential implies differentiability because it is inconsequential for our setting. What is actually important is that the subgradient is unique (which makes the first order oracle uniquely defined). Whenever we use $\nabla$ it is also justified (e.g. in the proof of eq 25: “This amounts to showing that $v_S^s$ is the *unique* maximizer…”)
>
> We will add further fundamental background in convex calculus.
>
> ---
>
> > **W4**
>
> GD is described in lines 196–203. Specifically at line 208 we explicitly state that it queries a subgradient (*not* an estimate) of the empirical risk $F_S$. If further clarifications are needed please let us know.
>
> ---
>
> > **W5**
>
> These further details can be added, but please notice that line 881 only claims that the *first* step does not involve projections (i.e. GD after one single step). This is not something that is hard to ensure.
> As we explicitly show $w_1=(\eta/m) v_S$ this only requires noticing that $\eta/\sqrt{m} <1$ as $\|v_S\|<\sqrt{m}$. This holds since $\eta<1$ and $m>80$.
>
> ---
>
> > **W6**
>
> Let us clarify. The generalization from one coordinate to multiple coordinates relies on the fact that changing additional coordinates can only decrease the first term in the maximum, while the second term already accounts for the worst-case increase over all coordinates. Thus, if modifying a single coordinate does not increase the overall expression, modifying multiple coordinates will not increase it either. We will clarify this argument in the revision.
>
> ---
>
> > **W7**
>
> We are a little bit confused here. f is defined in eq. 16 right above, and it is the only f that appears throughout the proof of Lemma A.1. It is indeed the sum of 5 and not 6 convex Lipschitz functions and we will correct this typo, but not two arguments, so clearly we are missing something here.
>
> In any case, if the reviewer can add further details, we are happy to clarify on every function why it is convex: Generally speaking, functions that are the sum of convex functions or maximum over convex functions (in particular linear terms) we consider as “easy to see” to be convex.
>
> ---
>
> > **W8**
>
> We are not sure what the reviewer means here. Claim 1 is contingent on a certain regime of parameters (thus it becomes trivial if the regime is infeasible), and when we apply it we choose explicitly parameters that fall in the regime. Consequently, our choice of parameters proves the regime is nonempty. We can add further details why our choice of parameters fall in the regime, but this follows direct and immediate calculations.
>
> ---
>
> > **W9**
>
> Thank you very much for the comment. We will reconsider the organization, incorporate suggested improvements and add further details in the proofs.
>
> ---
>
> > **Minor concerns**
>
> Regarding Eq. (8), the parameter $\eta$ is defined earlier in the GD paragraph as the step size (a few lines above this equation). We will make this more explicitly near Eq. (8).
> We will also clarify early in the paper what “linear dimension” means, namely that the ambient dimension scales linearly with the sample size, i.e., $d=\Theta(m)$.

---

> > ### Author Rebuttal · Reviewer_FUwg · 2026-04-01
> >
> > Thanks for taking the time to respond to my questions and concerns. Below I have a few remaining questions/concerns.
> >
> > W1+Q1: Please spell out the differences between this current paper and Guigues et al. (2017) in your response.
> >
> > W2+Q2: In author(s)' response, line 145: “we consider a function $f:W_d\times Z \to R$ that is convex and Lipschitz in its first argument”. This is not a rigorous statement. There are several issues. For example, is f(\cdot,z) convex and Lipschitz for all z, a.e. z, or in expectation? Is the Lipschitz constant invariant to different z? Depending on what kind of Lipschitz condition is imposed, even if f(\cdot,z) is convex on W_d, subgradient (which is repetitively used in the analysis) is not necessarily always available.
> >
> > W2+Q2 (continued): Please avoid overloading notation "f" to help improve readability.
> >
> > W3: While adding background discussions would help, I would recommend the authors to clarify the step of question using what the authors discussed in the rebuttal. A statement like: since subdifferential becomes a singleton, the gradient exists at the point of consideration. Then define the notation for the gradient.
> >
> > W4: The oracle statement that the authors point out is unclear or not rigorous. In particular, do you assume all subgradients are measurable?
> >
> > W6: Thanks for the effort in explanation. The description is a bit hard to understand precisely. Can you spell out this step more clearly and explicitly in the rebuttal?
> >
> > W7. The confusion was mostly caused by the author's typo of "6 convex Lipschitz" as well as notation overloading mentioned in W2+Q2. Current response is adequate.
> >
> > W8. Providing scenarios where the system of inequalities admit more explicit representations or examples of several feasible explicit choices of quantities could much help with understanding and evaluating the results there.
> >
> >  I am OK with the author(s)' current responses to the rest of my comments/questions.

---

> > > ### Author Response · Authors · 2026-04-02
> > >
> > > **W1+Q1**
> > >
> > > As we’ve tried to clarify in our response: Guigues, like Feldman, proves that there **exists** a bad erm, while in our result **all** erms are overfitting.
> > >
> > >
> > >
> > > **W2+Q2: In author(s)' response, line 145: “we consider a function  that is convex and Lipschitz in its first argument”.**
> > >
> > > The exact quote from line 145 is: “We assume that there exists a loss function $f :W_d\times Z\to R$ that is convex and **L**-Lipschitz in its first argument.” Respectfully, we cannot see how this can be interpreted in the wrong way. In particular, expectation doesn’t make sense here, especially as no distribution was introduced at this point. The Lipschitz constant is stated to be L, and there is no room for confusion that it depends on z.
> > >
> > > **W4:**
> > >
> > > Again, we are confused and not sure what is the so-called “Oracle Statement”, relevant textbooks that describe and explain the first-order oracle model are provided as references in the text. GD is depicted as an algorithm that receives a sample and returns a parameter. There is no distribution introduced here, so the question of measurability is unclear. On top of that, we explicitly restrict (without loss of generality) Z to be a finite space. Hence, no measurability issues appear in our final construction.
> > >
> > >
> > > **W6**
> > >
> > > We hope this clarifies the generalization of one-coordinate to all coordinates: Let i be any coordinate where $v$ and $v_S^s$ differ. Because v_S^s and w_t have the same sign in every coordinate, we have for every other v that: $v\cdot w_t= v_S^s\cdot w_t - (v_S^s -v)\cdot w_t = v_S^s\cdot w_t - 2\sum_{\{i': v_S^s(i')\ne v(i')\}} |w_t(i)| \le v_S^s\cdot w_t -2 |w_t(i)|$. We hope this clarifies.
> > >
> > > **W8. Providing scenarios where the system of inequalities admit more explicit representations or examples of several feasible explicit choices of quantities could much help with understanding and evaluating the results there.**
> > >
> > > These are the parameters we pick explicitly, and we hope this answers your question:
> > >
> > >
> > >  $\lambda^m=\frac{18}{\sqrt{2m}}$ , $\gamma^m=\frac{1}{m}-\frac{1}{\sqrt{2}m}$ , $\gamma^c=\min(\sqrt{\frac{\gamma^m}{200\sqrt{m}\eta T}},\frac{\lambda^c}{\sqrt{3}})$ and
> > > $\xi = \frac{\gamma^c}{\lambda^c}$

---

### Decision · Program_Chairs · 2026-04-30

**Decision:**

Accept (regular)

**Comment:**

The paper investigates the generalization properties of empirical risk minimization (ERM),  in stochastic convex optimization. The paper characterizes settings in which ERM incurs at least constant generalization error, establishes a different lower bound under stronger assumptions on the modulus of strong convexity and extend the analysis to gradient descent methods. The authors claim to resolve an open question posed by Feldman (2016) regarding the existence of linear-dimensional constructions where ERM yields unique yet overfitting solutions.

The paper provides a solution to an open problem indicates it significance. Most reviewers agreed that the paper merits acceptance, and after carefully reviewing the rebuttal and discussion, I share this view. That said, I recommend that the authors incorporate the reviewers’ feedback into the final version of the paper, with particular attention to:

1- The assumptions and claims are not discussed well enough

2- Some part the paper can be improved in term of presentation

3- Some related works are missing